# Contributions of advection and melting processes to the decline in sea ice in the Pacific sector of the Arctic Ocean

Haibo Bi[1,2,3,4], Qinghua Yang[5,6,7], Xi Liang[8], Liang Zhang[1,2,3,4], Yunhe Wang[1,2,3,4], Yu Liang[1,2,3,4], Haijun Huang[1,2,3,4]

[1]Key laboratory of Marine Geology and Environment, Institute of Oceanology, Chinese Academy of Sciences, Qingdao, China

[2]Laboratory for Marine Geology, Qingdao National Laboratory for Marine Science and Technology, Qingdao, China

[3]Center for Ocean Mega-Science, Chinese Academy of Sciences, Qingdao, China

[4]University of Chinese Academy of Sciences, Beijing, China

[5]Guangdong Province Key Laboratory for Climate Change and Natural Disaster Studies, and School of Atmospheric Sciences, Sun Yat-sen University, Zhuhai, China

[6]State Key Laboratory of Numerical Modeling for Atmospheric Sciences and Geophysical Fluid Dynamics, Institute of Atmospheric Physics (IAP),Chinese Academy of Sciences, Beijing, China

[7]Southern Marine Science and Engineering Guangdong Laboratory(Zhuhai), Zhuhai, China

[8]Key Laboratory of Research on Marine Hazards Forecasting, National Marine Environmental Forecasting Center, Beijing, China

*Correspondence to*: Haibo Bi (bhb@qdio.ac.cn)

## Abstract

The Pacific sector of the Arctic Ocean (PA, hereafter)  is a region sensitive to climate change. Given the alarming changes in sea ice cover during recent years, knowledge of sea ice loss with respect to ice advection and melting processes has become critical. With satellite-derived products from the National Snow and Ice Center (NSIDC), a 38-yr record (1979-2016) of the loss in sea ice area in summer within the Pacific-Arctic (PA) sector due to the two processes is obtained. The average sea ice outflow from the PA to the Atlantic-Arctic (AA) Ocean during the summer season (June-September) reaches $0.173 \times 10^6 \, km^2$, which corresponds to approximately 34% of the mean annual export (October to September). Over the investigated period, a positive trend of $0.004 \times 10^6 \, km^2/yr$ is also observed for the outflow field in summer. The mean estimate of sea ice retreat within the PA associated with summer melting is $1.66 \times 10^6 \, km^2$, with a positive trend of $0.053 \times 10^6 \, km^2/yr$. As a result, the increasing trends of ice retreat caused by outflow and melting together contribute to a stronger decrease in sea ice coverage within the PA ($0.057 \times 10^6 \, km^2/yr$) in summer. In percentage terms, the melting process accounts for 90.4% of the sea ice retreat in the PA in summer, whereas the remaining 9.6% is explained by the outflow process, on average. Moreover, our analysis suggests that the connections are relatively strong ($R = 0.63$), moderate ($R = -0.46$), and weak ($R = -0.24$) between retreat of sea ice and the winds associated with the Dipole Anomaly (DA), North Atlantic Oscillation (NAO), and Arctic Oscillation (AO), respectively. The DA participates by impacting both the advection ($R = 0.74$) and melting ($R = 0.55$) processes, whereas the NAO affects the melting process ($R = -0.46$).

# 1 Introduction

As the Arctic climate warms (Comiso, 2010; Overland et al., 2010; Graham et al., 2017), a wide range of researchers and the public show compelling interests on topics associated with the drop in sea ice coverage (Kay and Gettelman, 2009; Spreen et al., 2009; Polyakov et al., 2010; Zhang et al., 2010; Spreen et al., 2011; Woods et al., 2013; Tjernström et al., 2015; Notz and Stroeve, 2016; Screen and Francis, 2016; Koyama et al., 2017; Smedsrud et al., 2017; Stroeve et al., 2017; Niederdrenk and Notz, 2018). For the period since late 1970s, sea ice extent has been decreasing as revealed from a series of satellite microwave observations, ranging from -0.45%/yr to -0.39%/yr depending on different data products (Comiso et al., 2017). Specifically, the Arctic sea ice extent in September has been found to be the month with the most rapid decrease , at approximately -1.30%/yr for the period 1979-2017.

Comiso (2011) reported a more negative trend in the multiyear ice (MYI) coverage, approximately -1.72%/yr in winter over the period 1979-2011. As a result, the MYI extent decreased from approximately $6.2 \times 10^6$ $km^2$ in the 1980s to only $2.8 \times 10^6$ $km^2$ in the late 2000s. Recently, record lows in sea ice extent have been frequently set in summer over the past years (Serreze et al., 2003; Parkinson and Comiso, 2013; Stroeve et al., 2013). According to the National Snow and Ice Data Center (NSIDC) report (http://nsidc.org/yrrcticseaicenews/2012/09/), the Arctic sea ice extent plummeted to its minimum of $3.41 \times 10^6$ $km^2$ on 16 September 2012, approximately 18% below the previous record minimum in 2007. The significant decline in sea ice coverage in summer implies that less Arctic sea ice can survive the melting season to replenish the MYI cover in September (Kwok, 2007). Therefore, the Arctic Ocean is now dominated by younger and thinner first-year ice (FYI) (Maslanik et al., 2011; Tschudi et al., 2016), which is usually thinner and saltier and melts at a lower temperature.

Much attention is also paid to the decline in sea ice thickness in the Arctic Ocean. The sea ice extent covered by MYI in March decreased from approximately 75% (mid-1980s) to 45% (2011), and coverage fraction of the oldest ice (no less than fifth year) dropped from 50% of the multiyear ice cover to 10% (Comiso, 2011). Such a large decline in coverage of the thicker and older component of the MYI ice cover means a decrease in the mean ice thickness in the Arctic Ocean. For example, placing sea ice thickness derived from ICESat (2003-2008) in the context of a 43-yr submarine records (1958-2000), the overall mean winter thickness in a sizable portion of the central Arctic Ocean shows a decline of 1.75 m in thickness from 3.64 m in 1980 to 1.75 m from the ICESat record (Kwok and Rothrock, 2009). Moored sonars in the Fram Strait also observed a decrease in the annual mean sea ice thickness, from earlier 3.0 m (1990s) to recent 2.2 m (2008-2011) (Hansen et al., 2013). Due to Arctic sea ice age changes, satellite measurements from 2003-2008 (ICESat) and CyroSat-2 (2011-2015) reveal that there are net reductions in Arctic ice volume, of approximately $4.68 \times 10^3$ $km^3$ in autumn and $1.46 \times 10^3$ $km^3$ in winter (Bi et al., 2018). The thinner ice pack is more dynamically responsive to the drag forcing of winds and currents, allowing the Arctic sea ice to drift at a higher speed (Rampal et al., 2009; Spreen et al., 2011; Zhang et al., 2012; Kwok et al., 2013). Consequently, advective sea ice mass balance within the Arctic Ocean may have been changing.

Although we are familiar with the fact that the sea ice mass balance is closely related to the dynamic (advection) and thermodynamic (melting) processes, quantitative knowledge about their contributions to the sea ice area changes within the Arctic Ocean is scarce. Besides the utilization of satellite observations, modeling studies have been commonly used to diagnose the dynamic and thermodynamic forcing (e.g., Lindsay et al. (2009)). In this study, we examine the contributions of these two processes to the sea ice depletion within the PA side where sea ice loss is the most pronounced during summer (Cavalieri, 2012; Kawaguchi et al., 2014; Lynch et al., 2016; Comiso et al., 2017). In a preceding investigation, Kwok (2008a) examined the sea ice retreat within the PA sector due to melting and advection in summers of 2003-2007. However, the record of five years is too short to draw any robust conclusion about the variability and trend in the sea ice area changes due to the two processes. Kwok (2008b) investigated the sea ice transport between the PA and Atlantic-Arctic (AA) sectors for the period 1979-2007. However, the contribution of ice melting to sea ice decline in PA is beyond the scope of their study. Taking advantage of a longer and updated version of sea ice motion data provided by the NSIDC, this study attempts to quantify the contributions of the two processes (advection and melting) to the retreat of sea ice in summer within the PA sector over the period 1979-2016. Moreover, the possible causes for their variability and trends are examined by highlighting the role of large-scale atmospheric circulation.

We organize this paper as follows. The data and method are summarized in section 2. The estimates of sea ice outflow and melting are presented in section 3. The connection between sea ice retreat within the PA sector and typical large-scale atmospheric circulation is analyzed in section 4 by exploring the connection between the PA sea ice loss and the Arctic Oscillation (AO), North Atlantic Oscillation (NAO), and the Dipole Anomaly (DA). Section 5 reiterates the key findings and concludes this study.

## 2 Data and method

### 2.1 Sea ice motion and concentration

A gridded sea ice motion (SIM) product is provided by the National Snow and Ice Data Center (NSIDC) (http://nsidc.org/data/NSIDC-0116)(Tschudi et al., 2017a). SIM vectors are retrieved from a wide selection of platforms, from satellite radiometers, including the Advanced Very High Resolution Radiometer (AVHRR), Scanning Multichannel Microwave Radiometer (SMMR), Special Sensor Microwave Imager (SSM/I), Special Sensor Microwave Imager Sounder (SSMIS), and Advanced Microwave Scanning Radiometer-Earth observing system (AMSR-E), to the International Arctic Buoy Program (IABP) buoy data and surface winds from the National Centers for Environmental Prediction and the National Center for Atmospheric Research (NCEP/NCAR). The version of the SIM product (v3.0) is available with a grid of 25 km$\times$25 km, and is mapped on a EASE-Grid projection, which is a equal-area projection (Tschudi et al., 2017a; Tschudi et al., 2017b). The IABP buoy observation were employed throughout the product with a few unrealistic buoys being removed

due to errors (Szanyi et al., 2016). Likewise, AVHRR imagery is used for the period 1979-2000 with some error sources are excluded. Overall, the SIM data is estimated to has an uncertainty of between 1 and 2 cm/s depending on the amplitudes of both the sea ice concentration (SIC) and SIM (Sumata et al., 2015). At the time conducting this study, daily SIM data from 1979 through 2016 were available. Note that the newest version of SIM product (Tschudi et al., 2019) is soon released, based on which further analysis is expected.

Satellite-derived daily SIC records (1978-2017) (http://nsidc.org/data/NSIDC-0079) (Comiso, 2017) were also obtained from NSIDC. Common with SIM product, SIC is also extracted from multiple passive microwave observations from the SMMR, SSM/I, and SSMIS by the application of the bootstrap (BT) algorithm. Over the period November 1978 to July 1987 the ice concentration is available every other day. The data gap is filled using a temporal interpolation from the data of the two adjacent days (i.e., the previous and subsequent days). The concentration field utilized here is an up-to-date version (v3.1), offering improved consistency among the estimates from the different satellite observations through the application of varying tie points on daily basis. Furthermore, the product is optimized to further remove the effects of weather and land contamination (Cho et al., 1996). The data are available on a polar stereographic projection.

## 2.2 Procedure to estimate regional sea ice exchanges and melt

Following Kwok (2008a), the Arctic Ocean is divided into the PA and AA sectors (Figure 1). The division is defined by a line linking the easternmost tip of Severnaya Zemlya and the southwestern tip of Banks Island. With a length of 2840 km, the line serves as the gateway through which the sea ice area flux between the two sides of the Arctic Ocean is calculated.

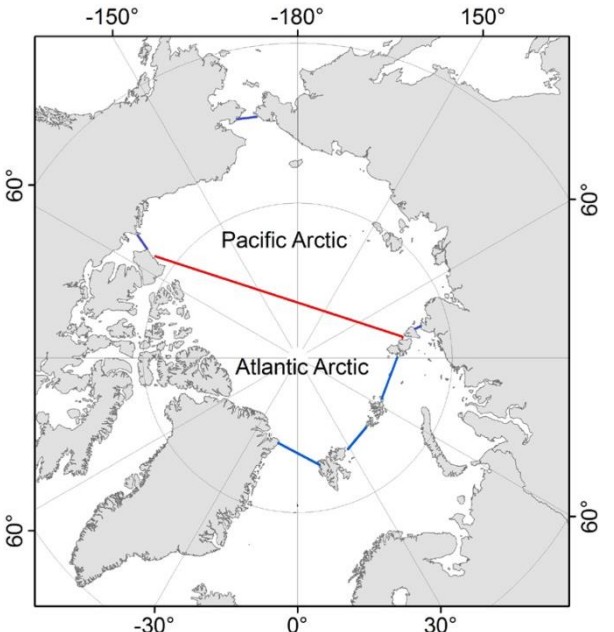

**Figure 1**. The fluxgate (red line) that is used to estimate sea ice area flux. The blue lines represents the boundaries of the PA and AA regimes. The endpoints in the North America and the Eurasia sides are (125.1 °W, 73.0 °N) and (101.0 °E, 79.6 °N), respectively.

Sea ice area flux is taken as the integral product of gate-perpendicular SIM and SIC for all grids across the fluxgate. The daily field of sea ice area flux ($F$, unit: km$^2$/day) is written as

$$F = \sum_{i=1}^{N-1} u_i c_i \Delta x \ (i = 1, 2, ..., N)$$

(1)

where $u$, $c$, and $\Delta x$ correspond to the gate-perpendicular SIM (unit: km/day), SIC, and the width of a grid (25 km) and $i$=1,2… $N$ refers to the index of grid cells along the gate. The length (unit: km) of the chosen gate corresponds to the total width of 113 grids. The monthly area flux ($F_m$) is the sum of daily fluxes for the corresponding calendar month, and the seasonal flux denotes the accumulative fluxes over the summer (June-September) and winter (October-May) months. Likewise, the annual flux is taken as the integrated flux over annual cycles between October and September (i.e. the sum of the winter and summer estimates). Although the focus season of this study is summer, the winter and annual fluxes across the gate are also presented for comparison to help readers understand the relative contribution of summer advection to the regional sea ice balance. We should note that the data quality for the daily SIC and SIM fields is likely lower during summer due to the surface melting process.

In our sign convention, sea ice transport from the PA to the AA sector is referred to as a positive flux (i.e., outflow) and the reverse direction is taken as a negative flux (i.e., inflow). Supposing that the errors in the SIM grid samples are unbiased, additive, uncorrelated, and normally distributed (Kwok, 2008b), the uncertainty of the daily fields can be expressed as follows.

$$\sigma_F = \frac{\sigma_d L}{\sqrt{N}}$$

(2)

where $L$ represents the fluxgate width (2840 km), and $\sigma_d$ is the uncertainty in the daily SIM. Regarding the daily drift uncertainty for one ice sample ($\sigma_d$), we use the uncertainty reported in Sumata et al. (2015), which is estimated by comparing NSIDC SIM data with buoy drifts and varies on the basis of the magnitudes of SIC and SIM. The uncertainty of the daily sea ice motion data during winter is obtained as 2 cm/s (i.e., 1.70 km/day) (Sumata et al., 2015). For the summer period, however, the ice motion is often blurred by surface melting. Therefore, summer ice motion mainly relies on interpolation that introduces additional uncertainty and thus has a poorer quality. As a result, we presumed summer-period uncertainty to be twice of winter error, up to 4 cm/s (i.e., 3.40 km/day) in the fluxgate. Assessment for ice concentration fields show there is a uncertainty of 3% in wintertime consolidated ice areas, whereas it is estimated to be about 5−10% in summer when

melt-ponding effects play a role (Meier, 2005). In this study, we use 3% and 10% as the representative value of ice concentration uncertainties during winter and summer periods, respectively (Meier, 2005).

The uncertainty of the seasonal ($\sigma_{summer}$ and $\sigma_{winter}$) or annual area flux estimates ($\sigma_{annual}$) can be described as

$$\sigma = \sigma_F \sqrt{N_D}$$

(3)

where $N_D$ denote the days for the period examined. Based on these equations, the expected uncertainties for the sea ice area flux estimates are obtained. The uncertainties listed in Table 1 correspond to 2.2%, 1.4%, and 1.2% of the mean flux (as shown in section 3.1.3) during summer, summer, and annual periods, respectively.

**Table 1.** Expected mean uncertainties in seasonal and annual sea ice area flux (unit: $10^6\,km^2$)

| Length of fluxgate | N | $\sigma_{summer}$ | $\sigma_{winter}$ | $\sigma_{annual}$ |
|---|---|---|---|---|
| 2840 km | 113 | 0.004 | 0.005 | 0.007 |

According to Kwok (2008a), the melting sea ice area in PA is the difference between the total sea ice area loss in PA and the sea ice area flux from PA to AA. A possible discrepancy is caused by deformation (divergence/convergence) processes, which can lead to reduction in sea ice area that may be misclassified as sea ice loss due to melt or export. The coarse resolution of satellite observations does not allow us to accurately quantify the sea ice loss in relation to sea ice deformation. However, in winter, sea ice area change due to deformation is expected to be negligible due to solid pack ice (approximately 1~2%) (Kwok et al., 1999). In summer, the deformation may be larger. A larger (smaller) convergence (divergence) is hypothesized north (south) of 80 N. For example, the accumulated divergence south (north) of 80 N is approximately 14% (25%) in 2007 (Kwok and Cunningham, 2012). The PA sector is an area located mostly south of 80 N where ice divergence is more likely to occur in summer, and new ice likely does not form within the divergence area due to warm temperature. In this study, it is difficult to quantitatively separate the ice loss due to deformation from the melting process. However, a sophisticated model study (Lindsay et al., 2009) suggests the convergence accounts for 1% of the Arctic basin ice loss in the Atlantic Sector. Since the Pacific sector is dominated by divergence over the major parts (Lindsay et al., 2009), convergence may contributes to a percentage no more than 1%. This is much less than the estimated melting trend, approximately 3.2%/yr for the 1979-2016 period as shown in Figure 11. Based on these findings, we ignore the limited contributions of convergence and divergence to sea ice area balance within the PA side.

**2.3 Large-scale atmospheric circulation index**

The atmosphere circulation modes screened in this study for possible linkages with sea ice area changes in PA include the AO (leading mode of sea level pressure (SLP) north of 20 N) (Thompson and Wallace, 1998), NAO (leading mode of SLP over the North Atlantic) (Hurrell, 1995), and DA (the second-leading mode of SLP within the Arctic Circle north of 70 N)

(Wu et al., 2005). Both the AO and NAO indexes (1979-2016) are available at the following sites affiliated with the Climate Prediction Center (CPC) at the National Oceanic and Atmospheric Administration (NOAA): http://www.cpc.ncep.noaa.gov/products/precip/CWlink/daily_ao_index/ao.shtml, and http://www.cpc.ncep.noaa.gov/products/precip/CWlink/pna/nao_index.html. The DA corresponds to the second-leading

mode of EOF of monthly mean sea level pressure (SLP) north of 70°N during the winter season (October mode o (Wu et al., 2005). The SLP fields are obtained from the National Centers for Environmental Prediction (NCEP) and the National Center for Atmospheric Research (NCAR). The record of the DA index (1979-2016) was provided by Bingyi Wu at Fudan University (personal communication).

## 2.4 Arctic climate variables

The changes in SLP can have significant impacts on winds ant hence SIM through their perturbations of other climate variables, such as surface air temperature (SAT), and precipitable water (PW). All these data are obtained from NCEP\NCAR in NOAA (Kalnay et al., 1996), with a grid size of $2.5°\times 2.5°$.

# 3 Results

## 3.1 Sea ice transport between the Pacific-Arctic and Atlantic-Arctic Ocean

**3.1.1 Comparison with a previous estimate**

To give credence to our estimates, we compared the estimates with the results reported by Kwok (2008b). He made use of the SIM data retrieved from the 37-GHz channels of the combined 29-yr SMMR and SSM/I time series between 1979 and 2007. Overall, the two estimates agree well with respect to summer and winter sea ice area fluxes (Figure 2). As shown in Figure 2, data pairs are distributed close to the Y=X line. There is a small mean positive bias between the two estimates, with

Kwok's estimates slightly larger than ours, approximately $0.015\times 10^6$ km$^2$ in summer (Figure 2a) and $0.028\times 10^6$ km$^2$ in winter (Figure 2b). These biases correspond to 8.6% and 8.2% of the mean estimates of sea ice area flux (as shown below in section 3.1.3) during summer ($0.173\times 10^6$ km$^2$) and winter ($0.337\times 10^6$ km$^2$) seasons, respectively. Moreover, a good consistency in terms of interannual variability is identified as indicated by high correlation coefficients ($R$ =0.91 in summer and $R$ = 0.96 in winter).

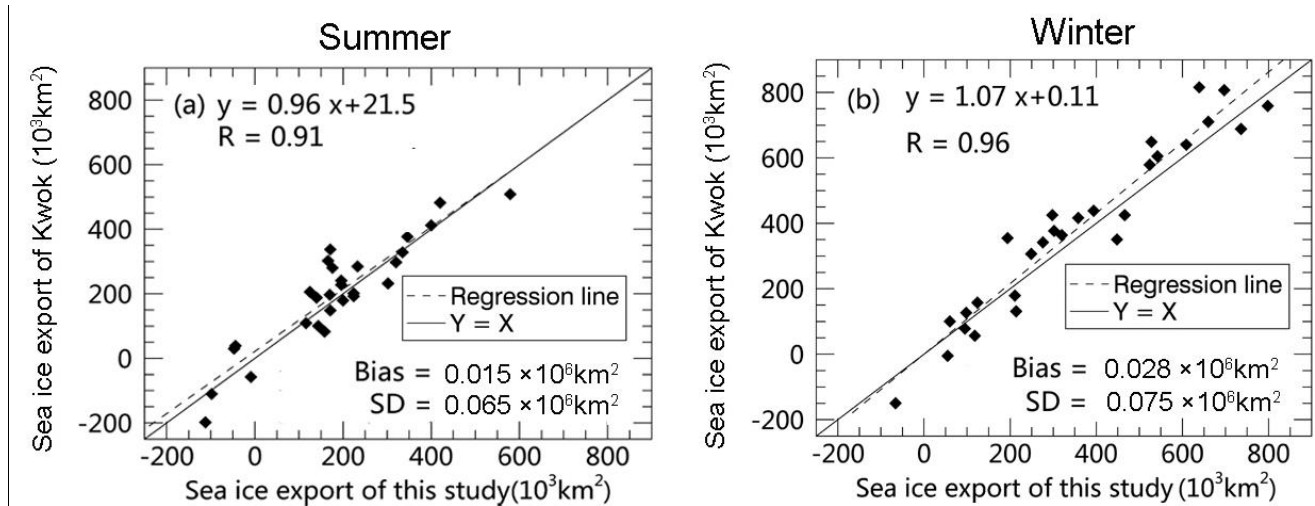

**Figure 2.** Comparison of regional sea ice exchanges between the PA and AA sectors in (a) summer and (b) winter. Our estimates are compared with a previous 29-yr record (1979-2007) provided by Kwok (2008b). The dashed line is the linear fit line between the two estimates. The solid line denotes for the 'Y=X' line. The linear relationship equation and correlation coefficient ($R$) between the two records, as well as the mean bias and standard deviation (SD) of the difference are also displayed.

### 3.1.2 Monthly sea ice area flux

The normalized monthly anomaly data have been adopted for studying the monthly variability. As a result, the seasonal variability is removed and, therefore, the distinct variations in monthly estimates are clearer and a direct comparison of variability among different months is feasible. The normalized or standardized procedure applied for the monthly anomaly fields ($F_a$) can be written as

$$F_a = \frac{F_m - F_m^{'}}{\sigma_{F_m}}$$

(4)

where $F_m$ represents monthly area flux and $F_m{'}$ and $\sigma_{F_m}$ indicate the means and standard deviations (SDs) of the corresponding month over the investigated period (1979-2016), respectively, and the results are shown in Figure 3.

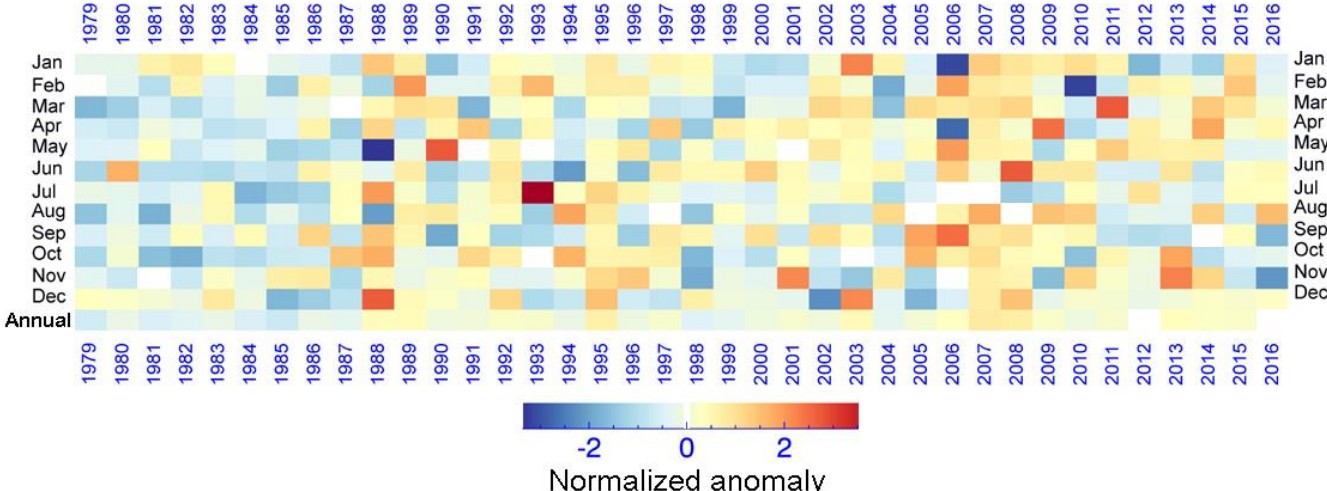

**Figure 3.** Normalized monthly anomaly fields of sea ice area flux. The anomaly field is calculated as the difference between the monthly estimate and the mean value of the same month computed over the period of interest. Then, the normalized anomaly field is obtained by dividing the monthly anomaly field by the standard deviation (SD) of the corresponding month over the 38-yr period. The bottom row denotes the annual mean value of the normalized anomaly.

Figure 3 displays the variations in anomaly fields for monthly sea ice flux between the two subsectors in the Arctic Ocean. Overall, the temporal variability is high (Figure 3). The first decade (1979-1988) is characterized by the occurrence of frequent negative anomaly fields, while for the remaining periods the emergence of positive anomalies seems to be more common (Figure 3). In particular, negative anomalies are observed in nearly all months (except for February) in 1985, yet almost all months experience positive ice flow anomalies in 1995, 2007 and 2008. The frequently-observed positive anomaly in recent periods compared to the early period (1979-1987) is further reflected in the annual mean normalized anomaly (bottom row in Figure 3).

The temporal variability in the monthly sea ice area flux fields is further emphasized in Figure 4, where the frequency distribution histogram for months with different normalized anomaly amplitudes are displayed for different decades (P1: 1979-1988; P2: 1989-1998; P3: 1999-2008, P4: 2009-2016). The dominance of anomalous low sea ice area fluxes during the first decade is reflected as an asymmetric distribution pattern in frequency (Figure 4a), with a larger number of months decreasing to the negative anomaly side. Comparatively, for the following three periods (Figure 4b-d), the distribution pattern begins to become more symmetric, mainly because of the growing number of months with positive anomaly fields.

To depict the decadal evolution in frequency distribution, the individual months for each period are binned into four groups with different anomaly amplitude ranges (A$\leq$ -1, -1<A$\leq$-0.5, 0.5<A$\leq$1, A > 1) are obtained and shown as inset text in Figure 4. The fraction is the ratio between the number of months affiliated with a specific range and the number of all months of the corresponding decadal period. In this case, there are 120 months for the each of the first three decades and 96 months for the last. Specifically, during the first decade (P1) approximately 43% of the months have normalized anomaly values less than -0.5, in contrast to only 15% of the months showing positive normalized values of greater than 0.5 (Figure

4a). In comparison, during the following three periods (Figure 4b-d), the fractions of months with significant negative normalized anomaly values less than -0.5 ( $-1<A\leqslant-0.5$ and $A\leqslant-1$) plummets to less than 28%, but 24~30% of the months are observed to have a distinct positive anomaly value at least of 0.5 (i.e. $0.5<A\leqslant1$ and $A>1$). In addition, the fraction changes for the extreme cases ($|A|\geqslant1$) remain relatively steady with time, varying between 26% and 29%. However, the extreme low anomalies ($A\leqslant-1$) reduce from 20.8% in P1 to 10.4%, while the extreme high anomalies ($A>1$) increases from 7.5% to 16.6%. This shift in sea ice exchanges between the PA and the AA sectors may indicate a shift of atmospheric circulation toward a pattern facilitating sea ice export out of the PA side (Wu et al., 2005; Zhang et al., 2008; Jia et al., 2009)

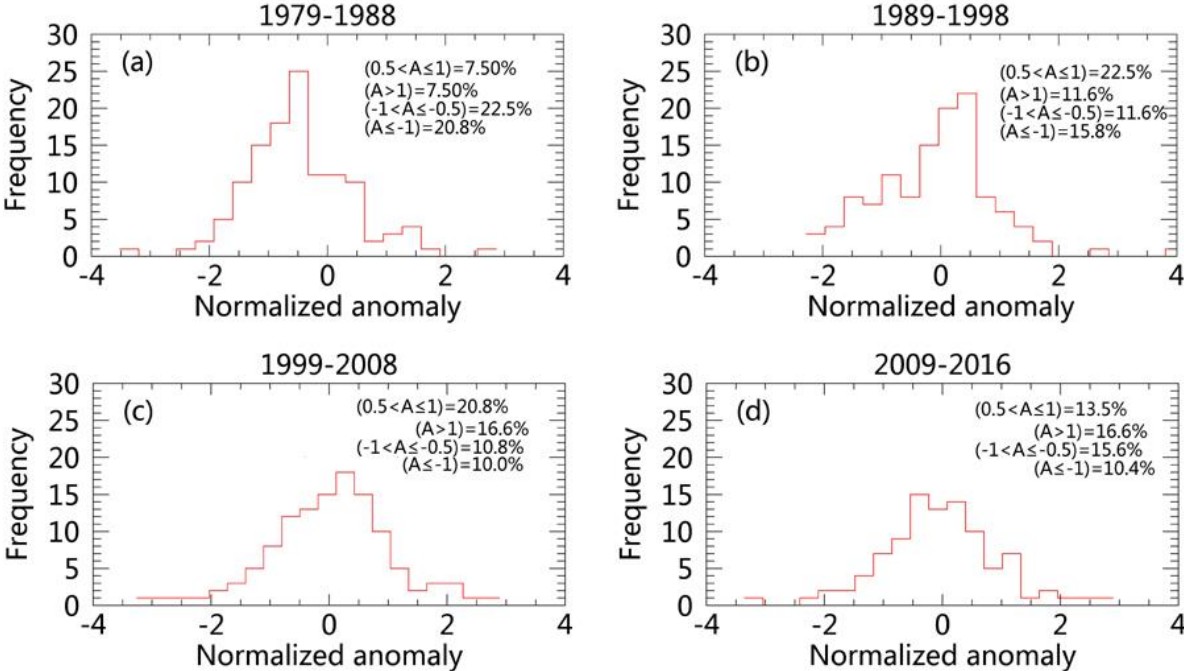

**Figure 4.** Frequency distribution histograms for months with different normalized anomalies in sea ice area flux in different decades. The fraction depicted as the percentage for each period, for example, $(0.5<A\leqslant1)$ = 7.5% as presented in panel (a), refers to the ratio between the frequency of the months sorted into the outlined range and the total number of months during the examined decades.

The monthly trends in the sea ice area flux are listed in Table 2. The trend in percentage for each month denotes the fraction of the monthly trend estimate relative to the mean sea ice area flux for the corresponding month. Basically, all months (except for January) show positive trends, with a mean value of 2.52%/yr. The largest trend appears in August (11.6% /yr, significant at the 90% level), whereas the lowest occurs in January (-3.67%/yr, not statistically significant). The positive trend in August is mainly associated with an increasing trend in SIM in the transpolar drift stream (Figure 5a). The negative trend in January is linked to the trend pattern from the Laptev Sea through the central Arctic to the east of the Beaufort Sea (Figure 5b). The changes in SIM explain a major part ($R^2$=0.98) of the trends and variability in sea ice area flux ($F_m$), and the remaining minor part is determined by the SIC changes which induce negative contributions to the monthly flux trends during summer but small positive contributions during winter months (December to April) (Table 2).

**Table 2.** Monthly trends for the sea ice area flux between PA and AA. The trends for the SIM and SIC fields over the fluxgate are also provided. (Unit: %/yr)

|  | Jan | Feb | Mar | Apr | May | Jun | Jul | Aug | Sep | Oct | Nov | Dec |
|---|---|---|---|---|---|---|---|---|---|---|---|---|
| $F_m$ | -3.67 | 0.07 | 5.28** | 3.27* | 2.68* | 1.53* | 1.17 | 11.6* | 1.77 | 4.32* | 1.77 | 0.43 |
| SIM | -3.66 | 0.35 | 5.49** | 3.67* | 2.94* | 2.04* | 0.53 | 10.44* | 2.23 | 3.95* | 2.47 | 1.20 |
| SIC | 0.007 | 0.016 | 0.010 | 0.009 | -0.033 | -0.072 | -0.165 | -0.531 | -0.650 | -0.183 | -0.002 | 0.012 |

Note:   * and ** denote the significance level at 95% and 99%, respectively.

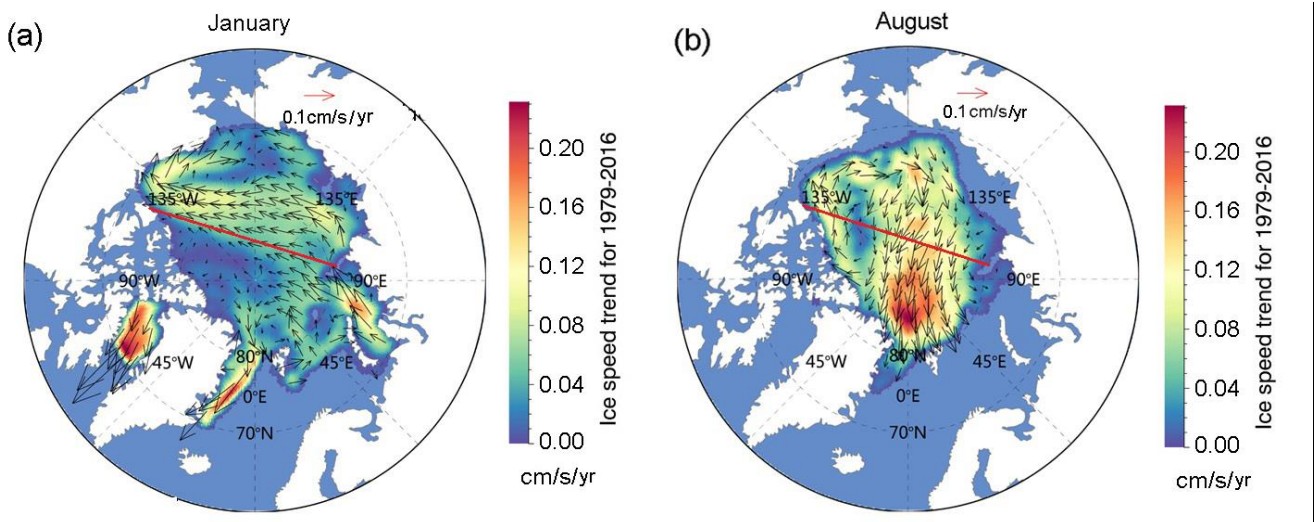

**Figure 5.** Sea ice motion trend in (a) January and (b) August. The magnitude of ice motion is indicated by background color and the length of arrows.

### 3.1.3 Annual and seasonal sea ice area flux

Figure 6 shows the estimates of the sea ice area transport for different seasons and years between the two sectors. The average annual sea ice export of 0.51 ($\pm$0.314)$\times 10^6$ km$^2$ comprises the mean winter and summer contributions of 0.337 ($\pm$0.263) $\times 10^6$ km$^2$ (or 66%) and 0.173 ($\pm$0.153) $\times 10^6$ km$^2$ (or 34%), respectively. The number after the sign "$\pm$" denotes the standard deviation of 38-yr sea ice area flux. Annually, sea ice area flux peaked at 1.089 $\times 10^6$ km$^2$ in 2007/2008 and set a record low -0.107 $\times 10^6$ km$^2$ in 1984/1985. Seasonally, the winter sea ice exports vary between -0.152 $\times 10^6$ km$^2$ (1998/1999) and 0.848 $\times 10^6$ km$^2$ (2013/2014), whereas the summer ice flux fluctuates within a range from -0.166$\times 10^6$ km$^2$ (1981) to 0.559 $\times 10^6$ km$^2$ (2006). The negative (positive) sea ice area flux as mentioned above points to a net sea ice inflow (outflow) from the AA to PA side (from the PA to AA side).

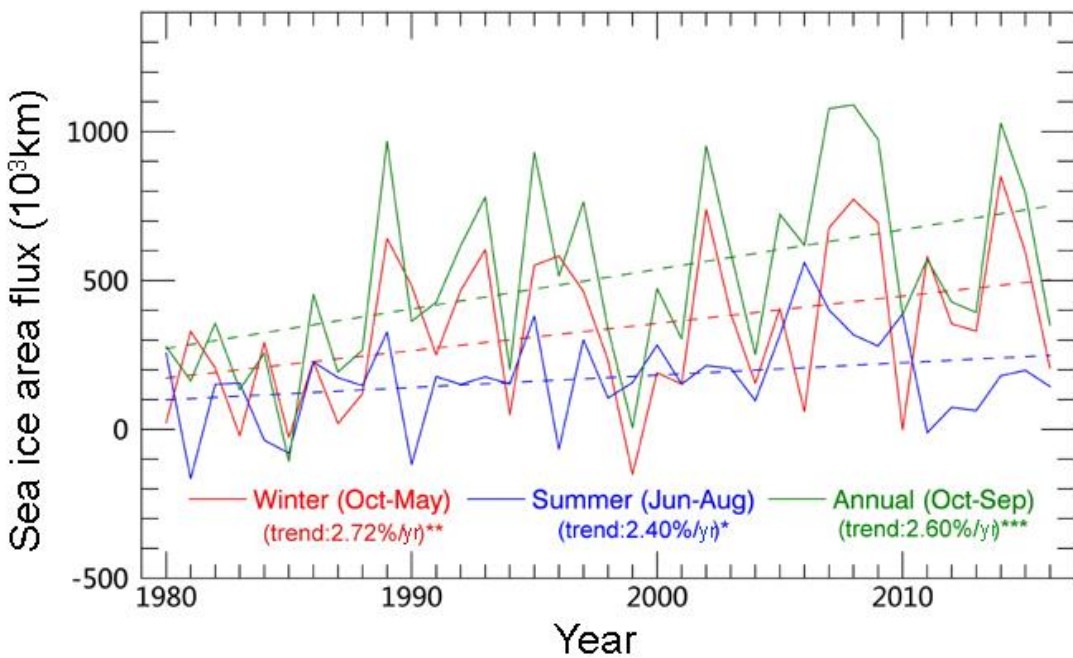

**Figure 6.** Annual and seasonal sea ice area fluxes between the PA and AA sectors over the period 1979-2016. The annual ice flow (annual cycle) is shown by the green line. Summer (from June through September) and winter (from October through May) fluxes are denoted with blue and red lines, respectively. The dashed lines representing the linearly fitted trends are obtained through the application of the least squares method. The labels *, **, and *** correspond to the significance at the level of 90%, 95% and 99%, respectively.

Over the 38-yr period, significant positive trends are observed for the sea ice area fluxes during the summer and winter seasons (Figure 6). Over the 38-yr period, the winter sea ice export exhibits a positive trend of $0.009 \times 10^6$ km$^2$/yr (i.e., 2.72%/yr, significant at a 95% level), and the summer export increases at a rate of $0.004 \times 10^3$ km$^2$/yr (i.e., 2.40%/yr, significant at a 90% level). Together, they contribute to an upward trend in annual sea ice area flux of $0.013 \times 10^6$ km$^2$/yr (i.e., 2.61%/yr, significant at a 99% level). The trend within the parentheses, expressed as percentage per year, is taken by dividing the trend estimate by the corresponding 38-yr mean estimate of sea ice area flux. The sharp increase since 1989 and onward for the winter export (Figure 6), in contrast to the preceding period 1979-1988, preconditions for a significant positive trend in the winter as well as annual fields of outflow. Nonetheless, the summer ice area flux features a gradual increase over the whole period, with anomalously large ice export occurring between 2006 and 2010 (approximately $0.310 \times 10^6$ km$^2$, on average), which is favorable for the maintenance of the overall positive trend in summer.

Due to the extensive coverage in longitudes and latitudes across the investigated fluxgate (2840 km), regional variations in the trend of the across-gate sea ice area flux fields are expected. Broadly, the Arctic sea ice circulation is characterized by the Beaufort Gyre (BG) and transpolar drift stream (TDS). How does the SIM trend vary in these two regimes? To answer this question, we not only present the overall pattern of spatial distribution of SIM trends over the entire Arctic Ocean (Figure 7), but also depict the details of the cross-gate SIM trends in fields (Figure 8). Figure 7 shows that SIM increases in the BG and

TDS regimes during both winter (Figure 7a) and summer (Figure 7b) seasons. In particular, the increasing SIM trends in the narrow southern arm of the BG regime appear in winter (Figure 7a) and summer (Figure 7b) The SIM trends during winter (summer) of approximately 0.12 cm/s/yr (0.11 cm/s/yr) (i.e., approximately 0.10 km/yr) over the southern arm of the BG lead to a reduced net sea ice export (i.e., more ice inflow) through the west end of the fluxgate (Figure 8a and b). The increasing SIM in the TDS is 0.07 cm/s/yr (0.04 cm/s/yr) (i.e., approximately 0.06 km/yr (0.03 km/yr)) during the winter (summer) season. Therefore, increasing sea ice outflow through the TDS is compensated by sea ice inflow associated with the BG. Compared with that in winter (Figure 7a or 8a), the sea ice trend in the TDS (Figures 7b and 8b) shifts its central axis to parallel the prime meridian. Overall, the SIM trend pattern appears to form an Arctic-wide anticyclonic mode. As discussed below, this mode has crucial meanings for the retreat of sea ice in the Pacific sector during summer.

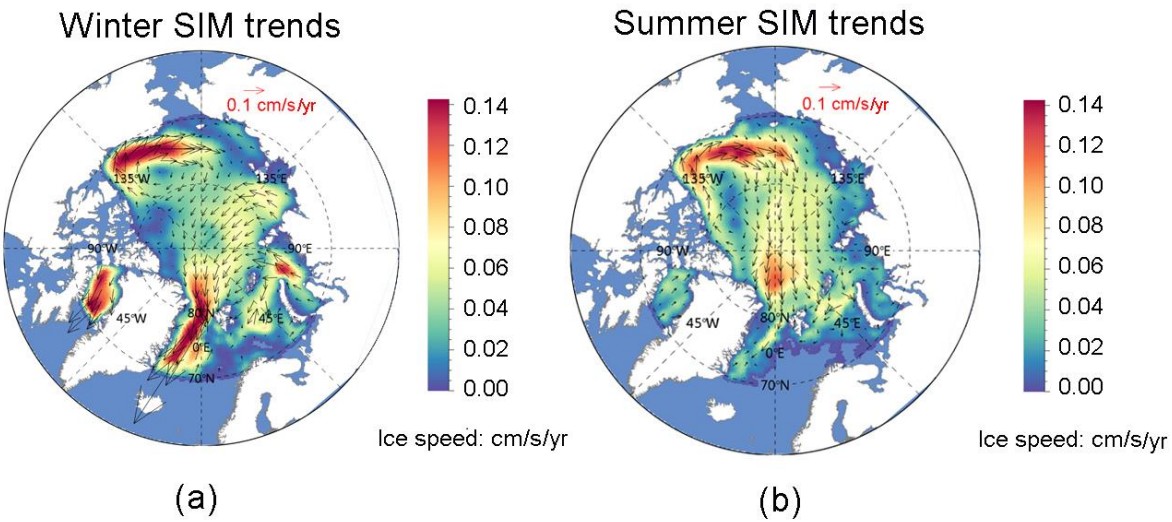

**Figure 7.** SIM trends during (a) winter (October- May) and (b) summer (June - September) over the period 1978/1979-2015/2016 and 1979-2016, respectively.

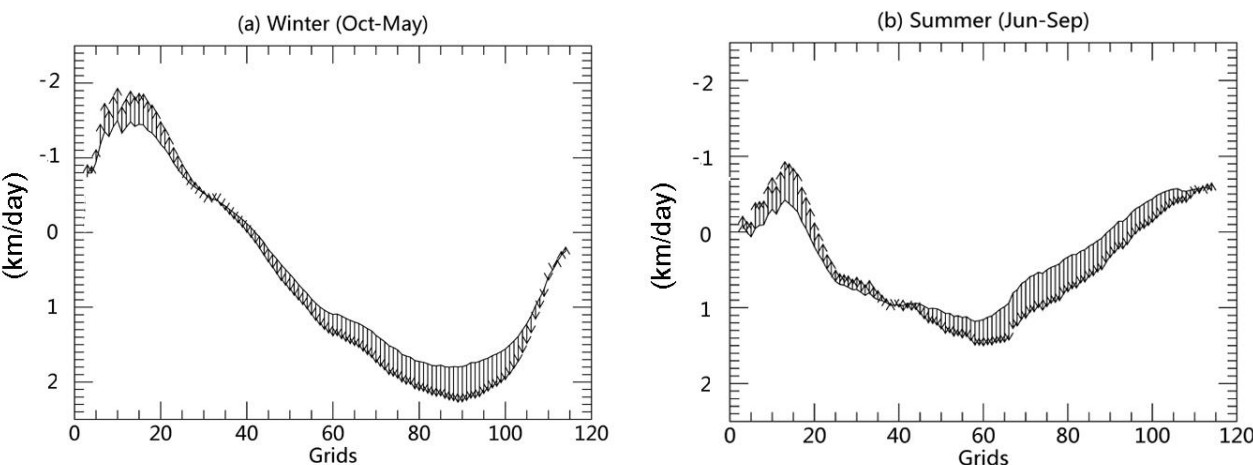

**Figure 8.** Cross-gate SIM climatology (black solid line) and trends (arrows) for (a) winter and (b) summer over the 38-yr period. The left end represents for the North America side and the right for the Eurasia side.

## 3. 2 Melting sea ice within the Pacific sector in summer

Sea ice melting is a sensitive indicator reflecting the warmer Arctic climate. An estimate of melting area of sea ice (dot-dashed line in Figure 9) is obtained as the difference between the observed sea ice area loss in the PA (red line in Figure 9) and the sea ice area flux through the fluxgate (blue line in Figure 9) during the summer season (June-September). Over the 38-yr period, the mean melting area is $1.66 \times 10^6 \, km^2$, with a distinct variation from $0.68 \times 10^6 \, km^2$ (1980) to $3.49 \times 10^6 km^2$ (2012). Noticeably, the PA sector seems to have been shifted into a new era from 2007 onward, with large ice melting of $2.70 \times 10^6 \, km^2/yr$ during the post-2007 period compared with $1.70 \times 10^6 \, km^2/yr$ for the period before 2007. In contrast, the sea ice area in AA sector during summer has scarcely changed before 2008 (the first three rows in Figure 9). AA is located in a higher latitudes where less melting is expected compared to that in PA. However, recent sea ice area changes reveal a shrinkage of sea ice area approximately $0.28 \times 10^6 \, km^2$ within the AA sector in September 2012 to 2013 (Figure 9). The sea ice loss within the AA may be more evident if the starting date of melting becomes earlier (Stroeve et al., 2014) and melting intensity is reinforced by the positive feedback mechanism of ice albedo (Perovich et al., 2007; Perovich et al., 2008; Screen and Simmonds, 2010) and/or if sea ice outflow through the Fram Strait is enhanced (Kwok et al., 2013; Bi et al., 2016; Smedsrud et al., 2017).

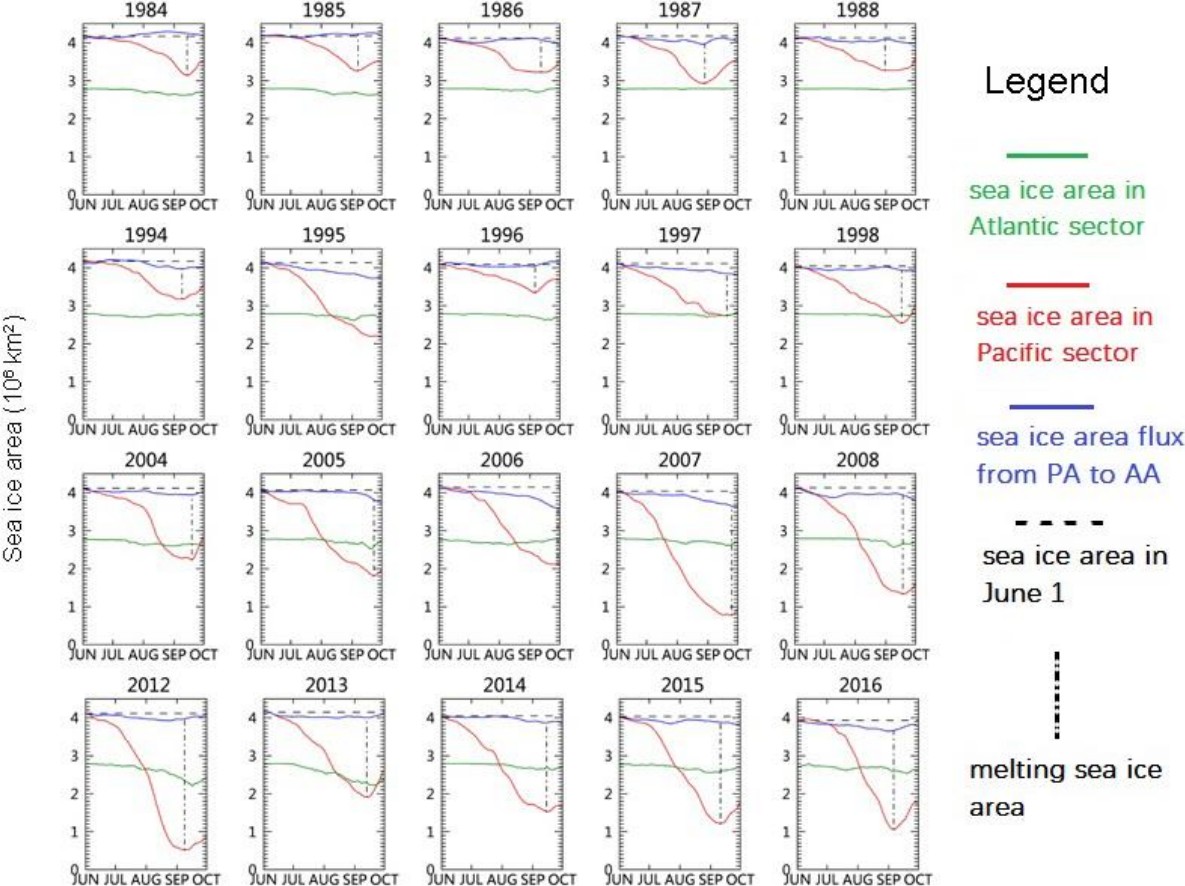

**Figure 9.** Selected daily changes in sea ice areas within the PA (red) and AA (green) during summer (June-September) for the period 1979-2016. The horizontal dashed line represents the sea ice area in the PA on 1 June, which is used as a

benchmark to measure the sea ice area changes due to melting. The cumulative daily sea ice area flux, with reference to the top dashed line, is shown as blue line. The melting sea ice area, denoted by the vertical dot-dash line, is taken as the difference between the total decline in sea ice area within the PA and the accumulated flux through the gate.

With the sea ice area budget shown in Figure 9, we quantify the relative contribution to sea ice area changes in summer due to the advection and melting processes over the period of 1979-2016 (i.e., Melting (M) = Observed area change (O) − Advection (A)). As a fractional contribution, the advection and melting processes, on average, account for 9.6% (i.e., A/O×100%) and 90.4% (i.e., M/O×100% or (A-O)/O×100%), respectively, of the observed ice area loss within the PA sector during summer months. Interannual variability in fractions is distinct, as suggested by the standard deviation of approximately 9%. However, no significant trend is identified in the fractions of sea ice area in connection with the two processes (Figure 10). The smallest (largest) fraction of 72.6% (110.2%) in the sea ice area change due to melting was observed in 2006 (1979), along with the largest (smallest) fraction of 27.4% (-10.2%) due to sea ice area advection (Figure 10). The negative fraction with respect to advection suggests a net inflow from the AA to PA side. In this case, the actual melting area (O-A) is thus larger than the directly observed area changes (O), causing a melting fraction greater than 100%. Over the examined period, the cases with a net increase in advective ice area took place before 1985.

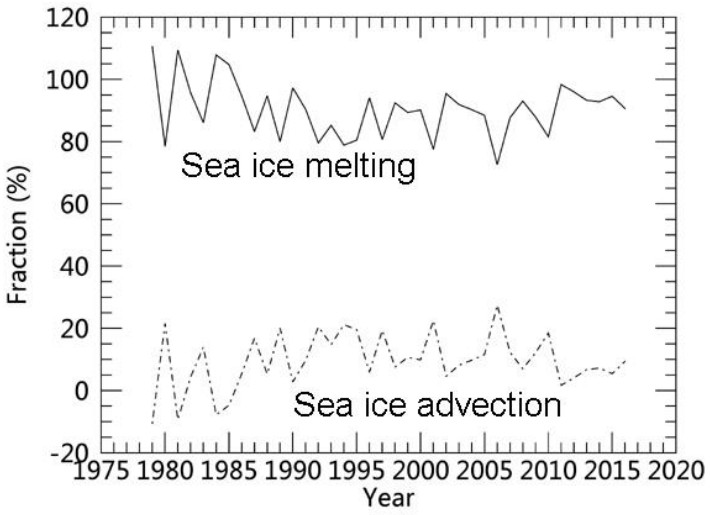

**Figure 10.** Fractions of sea ice area loss in the PA sector that are related to the sea ice melting and advection processes.

The trend of melting sea ice area within the PA is apparent over the 38-yr period (Figure 11), with an overall positive trend of 3.20%/yr (significant at the 99% level) (Figure 11). Since the mid 1990s, sea ice melting within the PA has been continually enhanced (Figure 11), which is associated with Arctic amplification (Screen and Simmonds, 2010). Decadal variability is also significant. The greatest increase of 7.48%/yr (significant at the 99% level) occurred in the third decade (1999-2008), whereas the remaining three decades show moderate or even negative trends.

Additionally, note that the day-of-year (DOY) of the minimum sea ice area in the PA sector displays an overall positive trend of 0.29 day/yr (Figure 12). The trend implies a gradually delayed occurrence of the DOY of the minimum sea ice area in the PA. There are also decadal variations in this DOY. In particular, over the second (1989-1998) and third decades (1999-2008), the trends in DOY approach to approximately 1.46 and 1.63 day/yr, respectively (Figure 12). The overdue

5    DOY is associated with the earlier beginning of melting date (Stroeve et al., 2014) and the positive sea ice albedo feedback loop (Perovich et al., 2008), which allows more heat to be absorbed in the area of ice loss, facilitates more melting of sea ice in the PA sector, and results in later occurrence of the minimum sea ice area. However, since 2009, the DOYs appear to have recovered to the former state in the 1980s (Figure 12). This reversion can be explained by the withdrawal of the sea ice extent toward a farther northern position at high latitudes (Figures 9 and 11), where freezing usually starts earlier in

10    comparison with areas at southern latitudes (Stroeve et al., 2014).

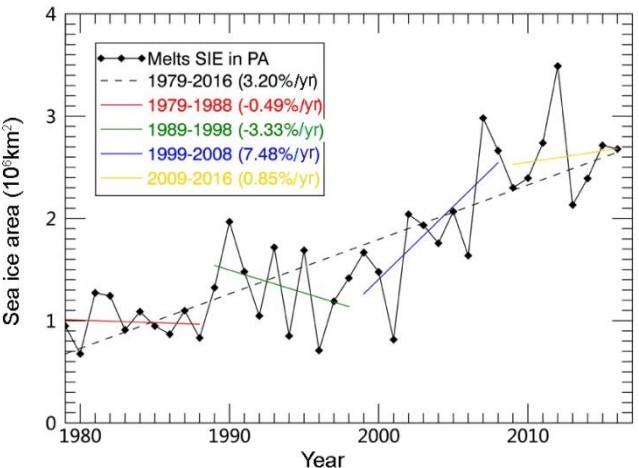

**Figure 11.** Time series for melting sea ice areas within PA

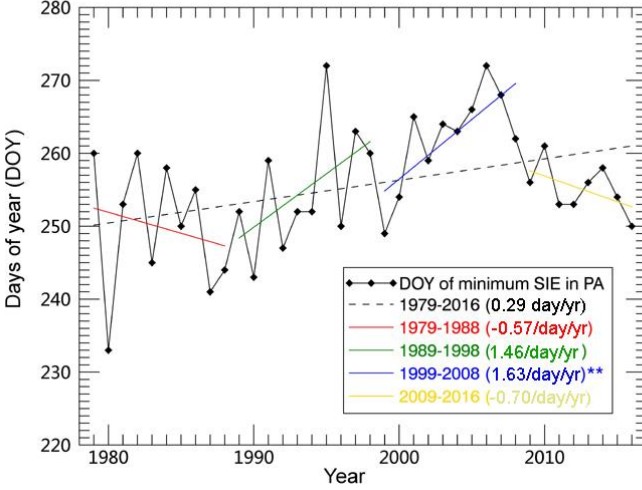

**Figure 12.** Day-of-year (DOY) of the annual minimum sea ice area in PA

## 4 Discussion

Wind forcing has been significant in modulating the sea ice variability in summer (Ogi et al., 2016). As an example, sea ice depletion induced by the summer melting process is connected to the wind forcing which can help to cause a warmer Arctic climate through the advection of warmer and moister air from the south (Wang et al., 2005; Zhang et al., 2013; Lee et al., 2017). The Arctic-wide wind forcing is linked to large-scale atmospheric circulation patterns (Zhang et al., 2008; Overland and Wang, 2010; Stroeve et al., 2011). Hence, the connection between the sea ice area loss in the PA sector and three typical atmospheric indices (AO, NAO, and DA) are assessed here. NAO and AO represent the dominant atmospheric circulation modes in guiding sea ice movement and Fram Strait export before 1994 (Rigor et al., 2002; Nakamura et al., 2015), while the DA seems to play a leading role over the latter period after 1995 (Wang et al., 2005). Here our objective is to examine how the sea ice variability due to advection and melting processes are quantitatively related to the interannual and decadal changes in these atmospheric modes. Furthermore, the potential impacts on sea ice loss in related to climatic variables (SAT, SLP, and PW) coupled with different atmospheric circulation pattern are highlighted.

Overall, the interannual variability for the three indices is large, and two indices, NAO and DA, reveal significant trends during summer (Figure 13). Over the investigated 38-yr period, the sea ice area reduction in summer within the PA seems to have been slightly influenced by AO fluctuations, with a low correlation coefficient ($R$) of -0.24 (Table 3). Separately, the sea ice area variations caused by advection and melting are barely attributable to the AO effects (Table 3). Therefore, in the following, we focus our analysis on the effects of the atmospheric circulations associated with NAO and DA, which show relatively strong correlations with the changes in sea ice area in the PA (Table 3).

**Table 3.** Correlations between summer mean atmospheric index and total sea ice area loss, and sea ice area decline due to melting and advection processes over the period 1979-2016.

| Atmospheric index (summer mean) | Melting sea ice | Net sea ice advection | Total ice retreat (Melting + advection) |
|:---:|:---:|:---:|:---:|
| AO | -0.28 | 0.14 | -0.24 |
| NAO | -0.47 | -0.18 | -0.46 |
| DA | 0.55 | 0.74 | 0.63 |

The influence of NAO-associated atmospheric circulation in winter on sea ice changes in summer has been broadly documented (Kwok, 2000; Jung and Hilmer, 2001; Parkinson, 2008), which is especially clear for the period from the late 1980s to early 1990s (Jung and Hilmer, 2001) when NAO was in its peak positive phase. In this study, the summer NAO is negatively correlated with the summer retreat of sea ice area within the PA sector (R = -0.46) over the 38-yr period. Table 2 shows a stronger connection with the sea ice area change due to melting (R = -0.47) than with the sea ice area loss in relation

to advection (R = -0.18). That is, a negative  NAO in summer would favor for more  sea ice melting and hence less ice coverage..

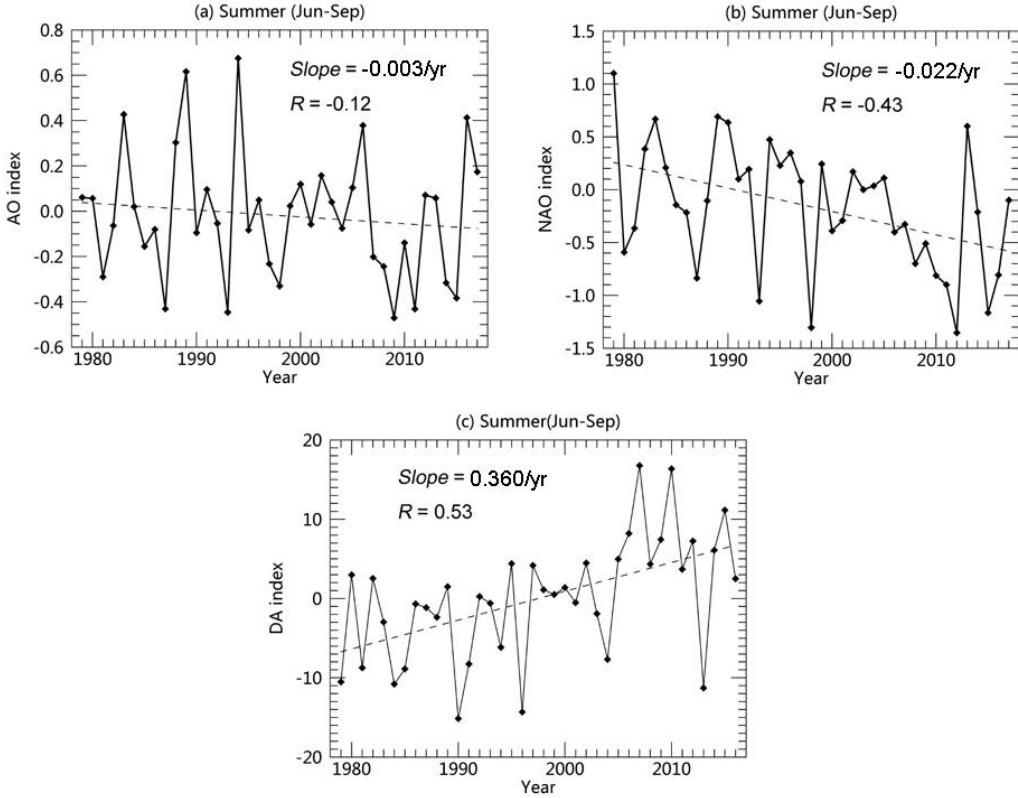

**Figure 13.** Variations in and trends of the mean atmospheric indices in summer (June-September), including (a) the AO, (b) the NAO, and (c) the DA index. The linear trends of NAO and DA indices are both significant the 95% significance levels according to the *t*-test method, while the AO index reveal an insignificant trend.

As there is a clear negative trend in the summer mean NAO index (Figure 13b), one may wonder how the summer NAO index trend is related to the increasing loss of sea ice within the PA sector. As shown in Figure 14g, the NAO-associated SLP distribution pattern, with a greater SLP in the western Arctic near the Canadian Archipelago, is favorable for a pattern of anticyclonic atmospheric circulation within the PA. Such a clockwise pattern of atmospheric circulation is hypothesized to push more ice from the eastern and northern Beaufort Sea to the western and southern Chukchi Sea (Figures 7b and 8b), where extensive sea ice melting has been commonly observed (Kwok, 2008a). The moderate negative correlation between NAO and melting sea ice area (R = -0.47) corroborates this hypothesis.

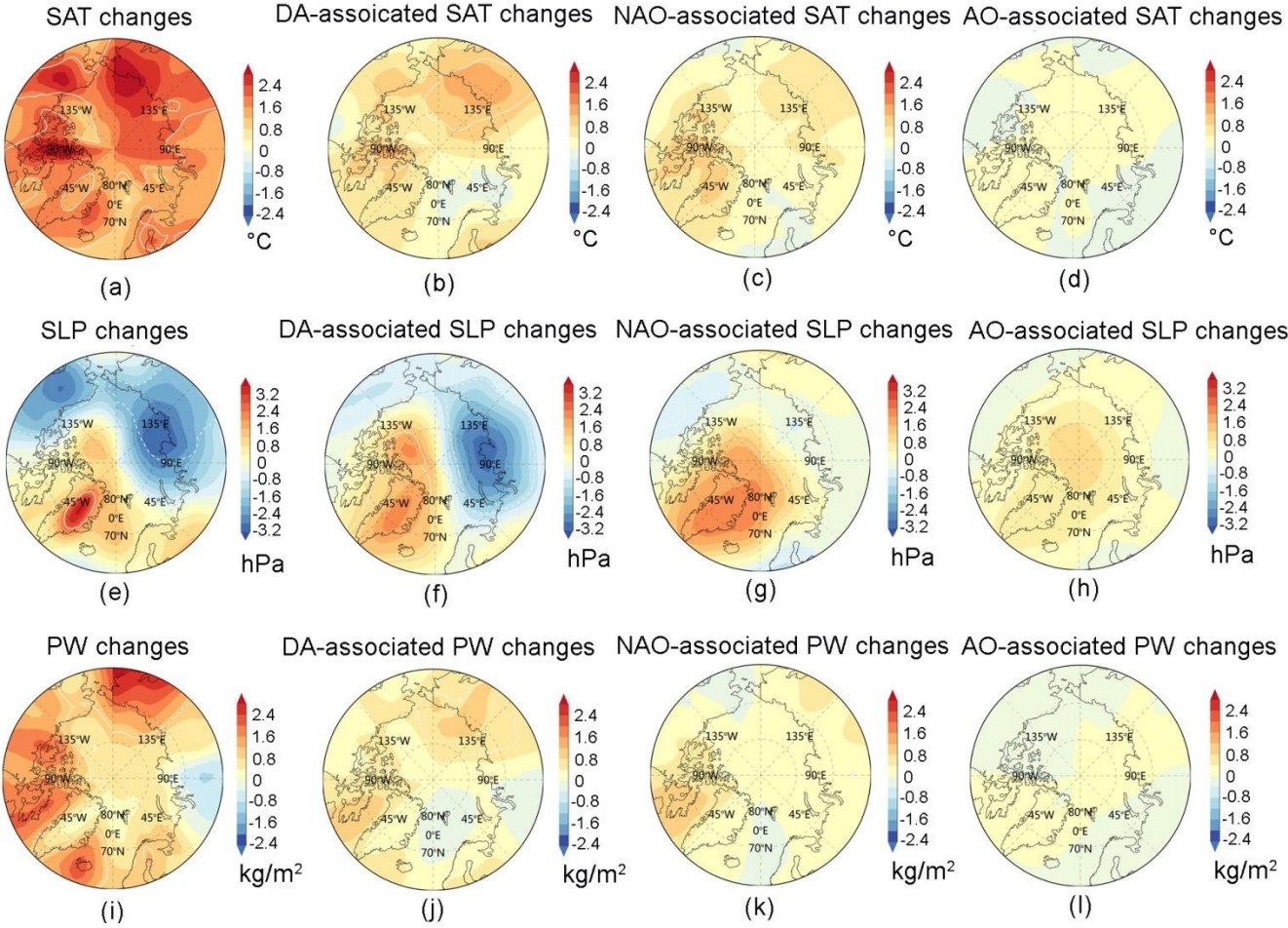

**Figure 14.** The changes in typical climate variables (SAT, SLP , and PW) over the 1979-2016 period (the first column, Figures a, d, and g). The change is obtained by multiplying the trend estimate (regression map) and the time span of the period (i.e., 38-yr). The regressions of these climatic variables on the DA (the second column, Figures b, e, and h) and NAO (the third column, Figures c, f, and i) are also presented.

The overall association between sea ice area variability and the DA index is comparatively robust (R = 0.63). Considering the clear positive trend in DA (1979-2016) (Figure 13c), the SLP changes explained by the DA trend are characterized by a dipole pattern with one action center over the Barents and Laptev Seas and the other center of action appearing on the opposite side over the Canadian Archipelago and Greenland (Figure 14f). In fact, the spatial distribution for the DA-associated SLP changes over the Arctic basin is broadly consistent with those for the actual SLP changes (Figure 14e), except for a prominent negative SLP center over the North Pacific and an enhanced positive SLP center over Greenland. With this type of SLP distribution, the DA-associated winds induce strengthened meridional ice advection through the TDS within the central Arctic (Figure 5, Figure 6b in section 3.1.3). A more rapid transpolar advection of sea ice is thus responsible for the observed increasing trend in the sea ice outflow from the PA to the AA sectors (Figure 7b). In addition, with such DA-associated SLP changes, more warmer air from the southern areas can be advected into and warm the SAT in the northern Arctic, finally augmenting the ice melting process (Zhang et al., 2013; Lee et al., 2017). Indeed, an SAT ridge

centered on the East Siberian Sea is observed (Figure 14b). As a response, the cold air of the high-latitude Arctic origin is push out toward the southern areas, reducing the local SAT over the eastern Greenland Sea, Barents Sea, and Kara Sea regions (Figure 14b).

Associated with the DA-associated SLP distribution pattern, the emergence of enhanced BG circulation is identified in the Beaufort Sea (Figure 14f). In particular, the sea ice drifts in the southern arm of the BG regime are largely strengthened (Figure 7b), which contributes to the large inflow of sea ice to the southern Beaufort Sea (Figure 8b), where sea ice undergoes dramatic melting processes in summer as indicated by the remarkable SAT increase there (Figure 14a). This coupled mechanism between dynamic (advection) and thermodynamic (melting) processes resembles that caused by the NAO. As a consequence, the depletions in sea ice in summer due to both melting and advection are relatively strongly correlated to the DA index of summer, with $R$ values of 0.55 and 0.74, respectively, over the entire period (Table 3).

Since AO shows a negligible trend (Figure 13a), the 38-yr climatic changes related to AO are insignificant (Figure 14d, h and i) and smaller in magnitudes compared to the DA- and NAO-associated changes. The NAO pattern is conventionally deemed as a regional index, representing parts of the broader AO pattern. However, NAO-associated SLP changes (Figure 14g) show a stronger gradient across the fluxgate than that of AO-associated SLP (Figure 14h), which would favors more sea ice outflow from PA to AA sectors. In comparison with NAO (Figure 14g), the AO-associated SLP distribution show a much weaker gradient across the Arctic Ocean (Figure 14h), although it may contributes to the sea ice advection from the Pacific side to the Atlantic side. As a result, lower correlations between AO and sea ice melting and advection processes are expected, with R = -0.28 and 0.14 (Table 3). However, these small overall correlations do not necessarily imply that AO plays no role in causing sea ice variations. For instance, throughout the examined periods, AO exerted more influences on sea ice changes for the earlier two decades (1979-1998), with R = 0.57 and 0.46 (Table 4).

The temporally varying association between atmospheric circulation and sea ice drift reflects Arctic climate changes. How does the linkage vary with time? Does it remains stable? To answer this questions, correlations between different summer mean atmospheric indices and minimum sea ice areas in the PA sector over different decades are obtained (Table 4). The AO effects are relatively unstable, imposing clear and decreasing positive effects over the first three decades but reversing to a negative moderate impact in the last period (P4) (Table 4). A clear association with DA arises during the latter two periods: P3 (1999-2008) and P4 (2009-2016). In contrast, the NAO index appears to have a significant impact only during the last period (2009-2016) (Table 4).

**Table 4.** Correlations between summer mean atmospheric index and sea ice minimum area in the PA sector for the different decades and the whole 38-yr period.

| Summer | P1 | P2 | P3 | P4 | Overall |
|---|---|---|---|---|---|

| (Jun-Sep) | 1979-1988 | 1989-1998 | 1999-2008 | 2009-2016 | 1979-2016 |
|---|---|---|---|---|---|
| AO | 0.57 | 0.46 | 0.39 | -0.34 | 0.24 |
| DA | -0.03 | -0.16 | -0.68 | -0.45 | -0.63 |
| NAO | 0.16 | 0.24 | 0.23 | 0.87 | 0.46 |

PW serves as an important indicator of Arctic climatic conditions. In view of the distribution of PW changes (Figure 14i), we find it broadly agrees with that of SAT in the PA sector (Figure 14a). For example, over the Siberian marginal seas, warmer SATs accompany more PWs. If the PW constituents drop to the surface as rain during a warming summer, they may benefit the melting process of the local ice/snow surface by lowering the surface albedo of ice/snow. Compared with the

NAO-associated changes (Figure 14k), the spatial distributions of DA-associated SAT (Figure 14b) and PW changes (Figure 14j) are, to a greater degree, consistent with those of the summer SIC trends (Figure 15), particularly over the area throughout the marginal seas in the PA sector, such as the Beaufort Sea, Chukchi Sea, East Siberian Sea, and Laptev Sea where the rates of sea ice area decline approach 2.0%/yr. On the other hand, we note that the magnitudes of SAT (or PW) changes in connection with the DA and NAO are far less comparable to the total SAT (or PW) changes, and similarity in

spatial distribution between Figure 14b and c (or Figure 14j and k) is not readily identifiable. This result occurs because aside from the effects of atmospheric circulation, other mechanisms also play important roles on the large and broad increases in SAT and thus PW over the Arctic, such as the sea ice albedo feedback loop, the intrusion of warmer Pacific/Atlantic Ocean water (Shimada et al., 2006; Tverberg et al., 2014; Alexeev et al., 2017), cloud feedback (Liu et al., 2008; Schweiger et al., 2008), and the changes in the strength in Atlantic meridional overturning current (AMOC) (Chen and

Tung, 2018).

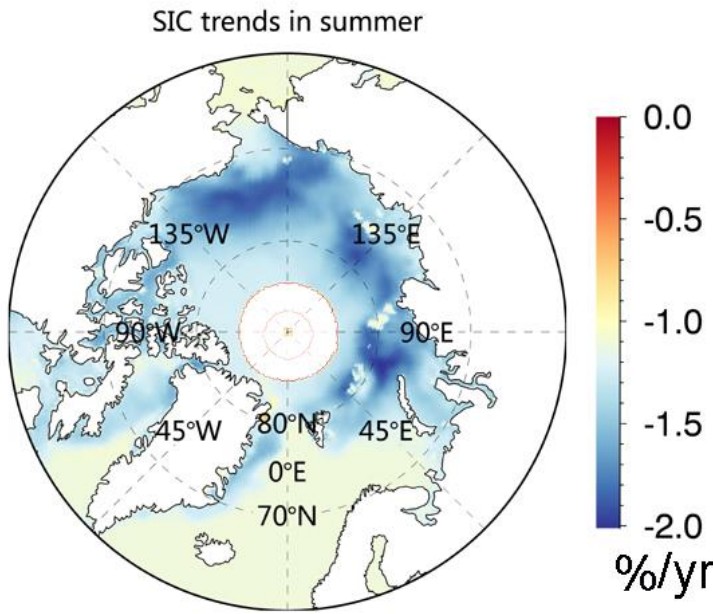

**Figure 15.** Summer SIC trends over the period 1979-2016

To summarize this section, we found different atmospheric forcing patterns exert varying influences on summer sea ice variability. Overall, the connections are relatively strong between sea ice loss and DA (R = 0.63), and NAO (-0.46), but fragile with AO (*R* = -0.24). In particular, the DA affects sea ice loss through both advection (*R* =0.74) and melting (*R* = 0.55) processes, whereas the NAO acts prominently on the melting process (*R* = -0.46). This is consistent with previous arguments that DA plays a more vital role in promoting summer sea ice depletion (Jia et al., 2009). Furthermore, our analysis shows that the connections are not invariant on decadal scale, but is characteristic with a regime shift (Table 4). Accordingly, the linkage between atmospheric forcing and sea ice loss is stronger during earlier two decades (1979-1998) with AO and NAO, but more robust with DA for the latter two decades (1999-2016). Since DA is becoming a more favorable pattern for summer sea ice loss through both dynamic and thermodynamic forcing, the causes for its surface emergence and amplification remains unclear and need further investigation.

## 5 Concluding remarks

Using the new version (v3.0) of NSIDC products (SIM and SIC), we quantify the contributions of the advection and melting processes to the sea ice retreat within the PA sector over the period 1979-2016. A synoptic view of their 38-yr variability and trends on different timescales is presented. Over this period, the annual (October to following September) mean sea ice export is 0.510 ($\pm$0.314)$\times 10^6$ km$^2$, with summer (June-September) and winter (October-May) outflows of 0.173$\times 10^6$km$^2$ (or 34%) and 0.337$\times 10^6$ km$^2$ (or 66%), respectively. A positive trend for the annual sea ice outflow is also identified, at approximately 0.013 $\times 10^6$ km$^2$/yr (or 2.08%/yr), which is attributable to the increasing export of 0.009$\times 10^6$ km$^2$/yr (or 2.72%/yr) in winter and increasing outflow of 0.004$\times 10^6$ km$^2$/yr (or 2.43%/yr) in summer. At the same period, sea ice area loss linked to the melting in summer, on average, amounts to 1.66$\times 10^6$km$^2$, with a significant positive trend of 0.053$\times 10^6$ km$^2$/yr (3.20%/yr). Further, we examine the relative roles of ice advection and melting in the decline of sea ice within the PA where dramatic sea ice retreat has occurred in summer during recent years. In percentage, the sea ice depletions in the PA due to the advection and melting processes account for 9.6% and 90.4%, respectively.

The linkage between sea ice loss and wind forcing associated with different large-scale atmospheric circulation during the summer season is explored. Overall, the AO is weakly connected to the sea ice loss on the PA sides, while the NAO is moderately correlated with the sea ice decline caused by melting. In contrast, the DA shows a more robust connection with the sea ice decrease in the PA through influence on both sea ice melting and advection. Dynamically, the DA-associated SLP conveys heat and moist air from the south to the north, resulting in the increasing SAT and PW over the marginal seas of the PA sector and contributing to the significant sea ice retreat. In addition, the positive trend in the DA induces stronger meridional wind forcing over the transpolar stream, leading to increased sea ice outflow and more ice decline in the PA. Thermodynamically, both the DA and NAO indices are associated with a strengthened anticyclonic SLP pattern over the

southern Beaufort Sea. This feature will promotes the westward transport of sea ice from the Canadian Basin to the south of the Beaufort Sea and Chukchi Sea where extensive melting sea ice has been detected in summers during the recent decade. By contrast, AO-associated sea ice changes due to melting and advection processes is not distinct, although temporally robust correlation is expected (Table 4).

The significant sea ice retreat also plays a crucial role in triggering regional responsive feedback in the atmosphere (Overland and Wang, 2010), as indicated by the warmer SAT (Figure 13a) and decreased SLP (Figure 13d) observed on the broad Siberian Arctic Ocean side. Therefore, if the current distinct sea ice loss on the Pacific-Arctic Ocean persists, the diminishing SLP in areas over the Laptev Sea, one of the centers of action of the DA, will further enhance the positive DA trend. Consequently, stronger dynamic (advection) and thermodynamic (melting) effects associated with the DA on sea ice

retreat within the PA sector are probably foreseen in the predictable future.

## Acknowledgements

We thank for the following organizations for providing the data used in this study. NSIDC provided the satellite-derived ice motion and concentration data, and the National Centers for Environmental Prediction/National Center for Atmospheric Research (NCEP/NCAR) provided the reanalysis product. Funds are provided by National Natural Science Foundation of

China under Grants 41406215 and 41706194, a fund provided by the Qingdao National Laboratory for Marine Science and Technology, and the NSFC-Shandong Joint Fund for Marine Science Research Centers (U1606401).

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
