# Peer review of "Contributions of advection and melting processes to the decline in sea ice in the Pacific sector of the Arctic Ocean"

_The Cryosphere, 2019_

## Referee Comment (RC1) · Anonymous Referee #1 · 1 Mar 2019

Review of "Contributions of advection and melting processes to the decline in sea ice in the Pacific sector of the Arctic Ocean" by Bi et al.

Summary

This paper presents an analysis of changes in sea ice in the Pacific sector of the Arctic. It notes the significant downward trend and delineates between advection and thermodynamic components. The decline is correlated to Arctic climate oscillations – the AO, NAO, and PDO. The strongest relationship is found with the DA, with a weaker correlation for the NAO, and the weakest for the AO. The NAO primarily is linked with thermodynamic processes (melting), while the DA link is strongest for advection, but is

also strong for melting.

General Comment

This is a nice paper that updates previous analyses of sea ice changes in the Pacific sector and the contributions of melt/growth and advection processes. The paper is well-written and thorough. The methodology appears sound and is explained well and the results are clearly presented. There are a few small questions/concerns, noted below. After addressing these, I find the paper acceptable after minor revisions.

Specific Comments (by page and line number):

P2, L4: "de" is non-standard and somewhat confusing for "decade". I would use either "dec", "d", or "decade".

P2, L11: what is the NSIDC report? Need to reference (if it's a webpage, I think it's fine to just include the link)

P2, L15: Could provide a more updated reference than Maslanik et al., (2011): Tschudi et al., Remote Sensing, 2016

P3, L17: Should provide the full citation (and include in the reference list) for the data, not just the website. NSIDC provides guidance on proper citation on their product website.

P3, L17: Note that a new version of the product will soon be released. At this point, I wouldn't expect your analysis to be redone and I don't think the new version would substantively change your results, but noting so that it is addressed in the final paper. Also, there is a new reference, submitted to The Cryosphere, documenting the changes:

Tschudi, M. A., Meier, W. N., and Stewart, J. S.: An enhancement to sea ice motion and age products, The Cryosphere Discuss., https://doi.org/10.5194/tc-2019-40, in review, 2019.

P3, L18-19: The first letters in the SSM/I, SSMIS, and AMSR-E written out sensor

names should be capitalized. P3, L22-23: There are a couple erroneous statements here. The SIM product is on the EASE-Grid, which is an equal-area projection – it is not polar stereographic. Also, some AVHRR images were removed due to errors, but AVHRR sources are still used for the 1979-2000 period. Likewise, a few buoys were removed, but buoys are used throughout the product. Maybe there just needs to be a rewording of the sentence to be clear.

P3, L27: As for the SIM, the full citation should be provided for the SIC product. NSIDC provides guidance on proper citation on their product website.

P4, L4-5: Again, an error here in the grid/projection info. The SIC is on a polar stereographic grid, but it is not equal area. I assume that this is just a miswording. However, if an equal area is assumed for the polar stereographic grid, that will give incorrect sea ice extent and area estimates. Each cell in the polar stereographic grid has a different area, so when summing for extent, you have to account for the specific area of each cell. If this was not done, then I would say that the analysis needs to be redone. NSIDC provides grid cell area files for the polar stereographic grids.

P5, L3: I can understand the focus on the summer, but it's important to note that the quality of both the SIC and SIM fields lower during summer due to surface melt water. I think reasonable estimates can still be obtained, but the lesser performance should be noted.

P5, L11: While the reference is provided, the uncertainty value is important and should be given explicitly here in the paper. A reader shouldn't' have to dig up a reference for what is salient information. And as noted above, summer uncertainties are higher.

P6, L10-11: What is the source for the DA index in terms of the data? I assume it's based on a reanalysis? That needs to be documented. Even if the values were provided directly by a colleague, the source that the colleague used needs to be cited.

P6, L13: Most importantly, SLP impacts winds and hence SIM.

P8, L2: I find the color scale for Figure 4 somewhat ambiguous – it's hard to tell where the 0 anomaly is exactly and which small values are positive or negative. I would suggest perhaps a gradient two-color anomaly scale with white denoting 0.

P9, L11: Earlier in the paper (e.g., in the abstract), "yr" is used, but here "a" (I assume for "annum"). You should be consistent in usage, so choose one or the other (I prefer "yr", but "a" is perfectly acceptable).

P10, 3: For Figure 6, it might make things too busy looking, but I think it could be helpful to overlay the flux gates. Especially for January, it's hard to tell if the vectors are advecting ice across the Pacific gate or not.

P11, L2: What are the "*"s next to the trends? I presume an indication of significance? That should be included in the caption.

P13, L4: The "summer season" should be defined here. It's noted further down on the page (June – September), but it should be specified when the term is first used.

P16, L10-14: I'm curious why AO shows such low correlation, but NAO shows higher. My understanding (which may not be complete or totally up-to-date) is that the two are very similar and that the NAO can be thought of as a regional expression of AO. Given that, I would've thought that AO would have higher correlation because it's hemisphere whereas the NAO is focused on the Atlantic sector. In other words, why would the Atlantic sector variability have much more effect on the Pacific sector than hemispheric variability?

P16, L23-25: This sentence is really difficult to follow. I'm not a big fan of the "positive (negative)" way of saying two things, but here it is especially tough because you use "positive (negative) NAO" for summer and then switch to "negative (positive) NAO" for winter. Trying to keep these straight is difficult! I would use "positive (negative) NAO" (or the reverse) for both cases and word the rest accordingly.

P17, L3: There is a lot of variability in yearly values, even when the correlation is fairly

high. Did you test for significance of the trends? I would guess that none of them may be significant. I think that would be good to include here and in the discussion.

P19, L23: Here again in Table 4, I'm curious about how different the NAO and AO correlations are. The differences are quite stark. In particularly, during the P4 period, the NAO and AO are even opposite signs. And while the AO has a higher correlation than the NAO through 2008, the NAO has had a much higher correlation since then. While earlier, you note that overall the AO correlation is weak and thus you focus on the NAO and DA, I think further discussion of the AO is warranted, particularly in terms of comparison with the NAO.

Technical Corrections

P4, L17: Use "length" instead of "distance".

P5, L28: typo, "does"

---

## Referee Comment (RC2) · Anonymous Referee #2 · 15 Mar 2019

Review of paper by Haibo Bi et al., "Contributions of advection and melting processes to the decline in sea ice in the Pacific sector of the Arctic Ocean," submitted for publication to The Cryosphere.

This is a solid but not a groundbreaking study. I recommend for publication with minor revisions.

Page 1, line 13 and elsewhere: Say "Pacific-Arctic sector of the Arctic Ocean" as in your title, not "Pacific-Arctic Ocean:" the latter does not exist. Same for "Atlantic-Arctic Ocean."

A major comment is the suggestion that you strive for consistency in your text and

graphics. IE: a) consistent units: i) Abstract uses 10ˆ3 kmˆ2 for PA -> AA outflow, and 10ˆ6 kmˆ2 for melting. Just use 10ˆ6 kmˆ2 for all, so that a reader can clearly compare eg 0.173 vs 1.66. ii) /yr vs /a: Sometimes you use "per year" and sometimes you use "per annum" and sometimes you do this in the same sentence! Just pick one and always use this. iii) /de vs /yr: I don't know if /de is a standard way to write "per decade" but in any case you mostly use /yr (or /a) so I suggest you translate the /de to /yr or /a. b) consistent terminology: Is the distance along your flux gate (2840 km) the WIDTH or the LENGTH? Sometimes you use one, sometimes the other. I think it should be the length; the width is 25 km ie the grid size, yes? c) consistent graphics scales: Figures 2 and 8: Use same vertical scale for left and right panels

Page 2, line 27: You could also include this reference for faster ice and changing drag, internal stress: Zhang et al. (GRL, 39, doi:10.1029/2012GL053545, 2012)

Page 2, line 31: There are a number of model studies that have considered dynamic vs thermodynamic forcings separately. One example is Figure 9 in Lindsay et al. (J. Clim., 22, doi: 10.1175/2008JCLI2521, 2009).

Page 3, lines 15-16: Just say "NSIDC." It is "University of Colorado" not Colorado University, but NSIDC is enough.

Page 3, lines 23-24: Please provide a reference for these unrealistic buoy velocities. Are you referring here to Szanyi et al. (Geophys. Res. Lett., 43, doi:10.1002/2016GL069799, 2016)?

Figure 1: a) Please give the lat/lon coordinates of each endpoint of the flux gate line. b) There is no need for the compass rose, and in fact it is inappropriate on a map with the North Pole included. c) Is there a southern boundary of the PA? maybe at Bering Strait? This should be noted with a blue line, just like for the southern AA boundaries. Equation 1: The units of F are lengthˆ2/time, yes? Please note this here.

Table 1: Please provide the units for ice area flux. Also, it's "length" not width, I think

(see comment above on consistency.)

Page 5, line 17: Your English is generally quite good, but there are numerous minor errors in spelling and grammar. EG here it is "annual" not "annul."

Bottom of page 5: Is it possible to provide a better estimate of the error in your calculations from the neglect of ice deformation (eg from numerical model studies)? Further, this error should be noted in the rest of your paper. EG if you find that there is a trend in total melt in the PA of X%, but this is smaller than the error from neglect of deformation, then do you have a significant result? Please discuss.

Page 6, lines 10-11: Is this DA index publicly available? Could you provide a link? AGU journals require a discussion of data availability; The Cryosphere may not, but it is all of our responsibility to discuss data availability so that our results can be independently reproduced.

Page 6, line 14: "precipitable"

Figure 3: It is not immediately obvious to me that there is more blue on the left and more red on the right. Some nice further analysis is provided in subsequent plots, but for this one, I might suggest the addition of the annual mean anomaly on a new bottom row, which could more clearly summarize the trend.

Page 9, line 2: This is very interesting. Any thoughts on why the extremes are not changing?

Figure 8: Use same vertical scale for both panels. Note that the line goes from N. America (left) to Eurasia (right).

Figure 9: It is nearly impossible for me to see these panels clearly and thus to interpret this figure. Could you try to make a better one, maybe with fewer panels?

Page 13, lines 6-9: Are you saying that melting in the AA is of similar magnitude to that in the PA, but that the AA ice is getting replenished by PA ice? This would be a major

new result if true. But I kind of doubt that it is true, given that the AA is (in the mean) farther north than the PA, and so probably there is more melt in the PA.

Section 4 Discussion re climate indices: The Arctic community went through a phase in which everything was correlated with climate indices. This fad has faded as ice continues to decline independently of climate indices. Further, climate indices don't provide predictive skill. It is unclear to me what they do provide, specifically in the present context of this paper. I might suggest that you write some introductory words to this section that explain why you are correlating your results with these indices, and then at the end, what your significant correlations provide in terms of new insight.

---

## Author Comment (AC1) · 9 Apr 2019

General Comment
This is a nice paper that updates previous analyses of sea ice changes in the Pacific sector and the contributions of melt/growth and advection processes. The paper is well-written and thorough. The methodology appears sound and is explained well and the results are clearly presented. There are a few small questions/concerns, noted below. After addressing these, I find the paper acceptable after minor revisions.
Response: We appreciate for the comments and suggestions and revise the manuscript accordingly.
Specific Comments (by page and line number):
P2, L4: "de" is non-standard and somewhat confusing for "decade". I would use either "dec", "d", or "decade".
Response: Based on this comment and suggestions from Reviewer 2# , we use unit "/yr" to take place of "dec" throughout the manuscript to ensure a unit consistency. Still many thanks for the reminding that "de" is not an appropriate word to be used in scientific paper.
P2, L11: what is the NSIDC report? Need to reference (if it's a webpage, I think it's fine to just include the link)
Response: The associated webpage link is added. (P2, L14 in the revised manuscript with a edit-tracking format. Hereafter, if not stated, the P and L correspond to the page and line numbers in the revised manuscript with a edit-tracking format)
P2, L15: Could provide a more updated reference than Maslanik et al., (2011): Tschudi et al., Remote Sensing, 2016
Response: The updated reference Tschudi et al., 2016 is added in the revised manuscript. (see P2, L16)
P3, L17: Should provide the full citation (and include in the reference list) for the data, not just the website. NSIDC provides guidance on proper citation on their product website.
Response: the relevant reference is added, see (Tschudi et al., 2017) in the revision.
P3, L17: Note that a new version of the product will soon be released. At this point, I wouldn't expect your analysis to be redone and I don't think the new version would substantively change your results, but noting so that it is addressed in the final paper. Also, there is a new reference, submitted to The Cryosphere, documenting the changes: Tschudi, M. A., Meier, W. N., and Stewart, J. S.: An enhancement to sea ice motion and age products, The Cryosphere Discuss., https://doi.org/10.5194/tc-2019-40, in review, 2019.
Response: I have to admitted that this is a timely and fantastic work since the NSIDC SIM and age products have been widely acknowledged in various studies. We found that the updated version 4 product is now (Access date: 2019-04-08) is in a process to be further improved based on some error reports. We keep in mind that if the newest version (v4.0) data is available, the final paper will use it for a further analysis. The associated reference by Tschudi (2019) introducing the new improvement regarding the product is added in the revised text (See P4, L6)

P3, L18-19: The first letters in the SSM/I, SSMIS, and AMSR-E written out sensor names should be capitalized.

Response: The first letters of the sensor name as referred is capitalized in the revision. (see P3, L26-29).

P3, L22-23: There are a couple erroneous statements here. The SIM product is on the EASE-Grid, which is an equal-area projection – it is not polar stereographic. Also, some AVHRR images were removed due to errors, but AVHRR sources are still used for the 1979-2000 period. Likewise, a few buoys were removed, but buoys are used throughout the product. Maybe there just needs to be a rewording of the sentence to be clear.

Response: Yes, the SIM product is available on the EASE-grid and we corrected accordingly. The sentences are reconstructed and reworded to make it clear that AVHRR and buoy measurements are used in the composition of SIM and some error sources have been excluded. Please see P4, L2-6.

P3, L27: As for the SIM, the full citation should be provided for the SIC product. NSIDC provides guidance on proper citation on their product website.

Response: the associated reference (Comiso, 2017) is added. (See P4, L11)

P4, L4-5: Again, an error here in the grid/projection info. The SIC is on a polar stereographic grid, but it is not equal area. I assume that this is just a miswording. However, if an equal area is assumed for the polar stereographic grid, that will give incorrect sea ice extent and area estimates. Each cell in the polar stereographic grid has a different area, so when summing for extent, you have to account for the specific area of each cell. If this was not done, then I would say that the analysis needs to be redone. NSIDC provides grid cell area files for the polar stereographic grids.

Response: The misleading information is corrected and the SIC projection is presented as a polar stereographic grid. See the concerning texts in the revision (P4, L18).

P5, L3: I can understand the focus on the summer, but it's important to note that the quality of both the SIC and SIM fields lower during summer due to surface melt water. I think reasonable estimates can still be obtained, but the lesser performance should be noted.

Response: This is true. Surface melting causes ambiguous observational signals which are really a big problem to retrieve a real sea ice motion or sea ice concentration. Following the suggestion, we remind the author that summer performance may be less good as that in winter. (Please refer to P5, L15-16)

P5, L11: While the reference is provided, the uncertainty value is important and should be given explicitly here in the paper. A reader shouldn't' have to dig up a reference for what is salient information. And as noted above, summer uncertainties are higher.

Response: The specific uncertainty values are summarized and explained in the revised manuscript. For example, the uncertainty of the daily sea ice motion data during winter is obtained as 2 cm/s (i.e., 1.70 km/day) (Ivanova et al., 2014; Sumata et al., 2015) (Sumata et al., 2015). Also, the uncertainty in summer SIM data is presumed to larger than that that used in winter. For more detail, please refer to P6,

L4-11.

P6, L10-11: What is the source for the DA index in terms of the data? I assume it's based on a reanalysis? That needs to be documented. Even if the values were provided directly by a colleague, the source that the colleague used needs to be cited.

Response: DA is retrieved from NCEP reanalyzed SLP product. The concerning information about DA is provided in the revision and a reference by Wu et al. (2005) is given for more detail about how the DA index is obtained. (See P7, L11-19)

P8, L2: I find the color scale for Figure 4 somewhat ambiguous – it's hard to tell where the 0 anomaly is exactly and which small values are positive or negative. I would suggest perhaps a gradient two-color anomaly scale with white denoting 0.

Response: We redraw the Figure 4 with a white color pointing to zero. (See Figure 4, for example, Feb 1979). Also, following suggestions by Reviewer 2, we added the annual average of the anomaly in the bottom row of the Figure.

P9, L11: Earlier in the paper (e.g., in the abstract), "yr" is used, but here "a" (I assume for "annum"). You should be consistent in usage, so choose one or the other (I prefer "yr", but "a" is perfectly acceptable).

Response: For consistency in the use of time, "yr" is used through the paper.

P10, 3: For Figure 5, it might make things too busy looking, but I think it could be helpful to overlay the flux gates. Especially for January, it's hard to tell if the vectors are advecting ice across the Pacific gate or not.

Response: For a clear look about the sea ice movement around the fluxgate, the gate line is overlaid on Figure 5.

P11, L2: What are the "*"s next to the trends? I presume an indication of significance? That should be included in the caption.

Response: Yes, the label "*" denotes for significance. '*', '*,*' , '*,*,*' correspond to the significance level at 90%, 95%, 99%. This notion is given in the caption of Figure 6 (P12, L11-12).

P13, L4: The "summer season" should be defined here. It's noted further down on the page (June – September), but it should be specified when the term is first used.

Response: The definition of summer is presented in P14, L7.

P16, L10-14: I'm curious why AO shows such low correlation, but NAO shows higher. My understanding (which may not be complete or totally up-to-date) is that the two are very similar and that the NAO can be thought of as a regional expression of AO. Given that, I would've thought that AO would have higher correlation because it's hemisphere whereas the NAO is focused on the Atlantic sector. In other words, why would the Atlantic sector variability have much more effect on the Pacific sector than hemispheric variability?

Response: The relevant explanations have been given in P21, L11-19. In addition, for a clear analysis, climatic variables (SLP, SAT, and PW) related to AO trending changes are presented in Figure 14 (see Figure 14d, h, and i).

Since AO shows a negligible trend (Figure 13a), the 38-yr climatic changes related to AO are insignificant (Figure 14d, h and i) and smaller in magnitudes compared to the DA and NAO-associated changes. The NAO pattern is conventionally deemed as a regional index, representing parts of the broader AO pattern. However,

NAO-associated SLP changes (Figure 14g) show a stronger gradient across the fluxgate than that of AO-associated SLP (Figure 14h), which would favors more sea ice outflow from PA to AA sectors. In comparison with NAO (Figure 14g), the AO-associated SLP distribution show a much weaker gradient across the Arctic Ocean (Figure 14h), although it may contributes to the sea ice advection from the Pacific side to the Atlantic side. As a result, lower correlations between AO and sea ice melting and advection processes are expected, with R = -0.28 and 0.14 (Table 3). However, these small overall correlations do not necessarily imply that AO plays no role in causing sea ice variations. For instance, throughout the examined periods, AO exerted more influences on sea ice changes for the earlier two decades (1979-1998), with R = 0.57 and 0.46 (Table 4).

P16, L23-25: This sentence is really difficult to follow. I'm not a big fan of the "positive (negative)" way of saying two things, but here it is especially tough because you use "positive (negative) NAO" for summer and then switch to "negative (positive) NAO" for winter. Trying to keep these straight is difficult! I would use "positive (negative) NAO" (or the reverse) for both cases and word the rest accordingly.

Response: The sentence is reconstructed for a clear expression. (P19, L4-6)

P17, L3: There is a lot of variability in yearly values, even when the correlation is fairly high. Did you test for significance of the trends? I would guess that none of them may be significant. I think that would be good to include here and in the discussion.

Response: The linear trends of NAO and DA indices are significant the 95% significance levels according to the t-test method, while the AO index reveal an insignificant trend. This is noted in the caption of Figure 13. (P19, L9-10).

P19, L23: Here again in Table 4, I'm curious about how different the NAO and AO correlations are. The differences are quite stark. In particularly, during the P4 period, the NAO and AO are even opposite signs. And while the AO has a higher correlation than the NAO through 2008, the NAO has had a much higher correlation since then. While earlier, you note that overall the AO correlation is weak and thus you focus on the NAO and DA, I think further discussion of the AO is warranted, particularly in terms of comparison with the NAO.

Response: The relevant explanations have been given in P21, L11-19. In addition, for a clear analysis, climatic variables (SLP, SAT, and PW) related to AO trending changes are presented in Figure 14 (see Figure 14d, h, and i).

Technical Corrections

P4, L17: Use "length" instead of "distance".

Response: Corrected as suggested (P5, L9).

P5, L28: typo, "does"

Response: Corrected as suggested (P6, L26).

---

## Author Comment (AC2) · 9 Apr 2019

Review of paper by Haibo Bi et al., "Contributions of advection and melting processes to the decline in sea ice in the Pacific sector of the Arctic Ocean," submitted for publication to The Cryosphere. This is a solid but not a groundbreaking study. I recommend for publication with minor revisions.

Page 1, line 13 and elsewhere: Say "Pacific-Arctic sector of the Arctic Ocean" as in your title, not "Pacific-Arctic Ocean:" the latter does not exist. Same for "Atlantic-Arctic Ocean." A major comment is the suggestion that you strive for consistency in your text and graphics. IE: a) consistent units: i) Abstract uses $10^3$ $km^2$ for PA -> AA outflow, and $10^6$ $km^2$ for melting. Just use $10^6$ $km^2$ for all, so that a reader can clearly compare eg 0.173 vs 1.66. ii) /yr vs /a: Sometimes you use "per year" and sometimes you use "per annum" and sometimes you do this in the same sentence! Just pick one and always use this. iii) /de vs /yr: I don't know if /de is a standard way to write "per decade" but in any case you mostly use /yr (or /a) so I suggest you translate the /de to /yr or /a. b) consistent terminology: Is the distance along your flux gate (2840 km) the WIDTH or the LENGTH? Sometimes you use one, sometimes the other. I think it should be the length; the width is 25 km ie the grid size, yes? c) consistent graphics scales: Figures 2 and 8: Use same vertical scale for left and right panels

Response: a) consistent units are used in the revised manuscript. i) we use $10^6$ $km^2$ for outflow and melting sea ice area. ii) we use /yr throughout the paper. iii) for the trends where '/de' is used, we change them to '/yr'. b) Following the suggestion, we keep using LENGTH for the distance spanned by the fluxgate. The grid size is 25 km. c) the vertical scale for the left and right panels in Figures 2 and 8 are changed to a consistent scale. The figures are redraw accordingly.

Page 2, line 27: You could also include this reference for faster ice and changing drag, internal stress: Zhang et al. (GRL, 39, doi:10.1029/2012GL053545, 2012)

Response: this is an excellent reference and we add it (P3, L1 in the revised version of the manuscript with a edit-tracking format). Hereafter, without specific statement, the page and line numbers denote those appear in the revision with a edit-tracking format.

Page 2, line 31: There are a number of model studies that have considered dynamic vs thermodynamic forcings separately. One example is Figure 9 in Lindsay et al. (J. Clim., 22, doi: 10.1175/2008JCLI2521, 2009).

Response: We carefully read this paper and found Lindsay's work is very interesting and of great relevance to our study. The paper is noted in the revised manuscript (P3, L5-6).

Page 3, lines 15-16: Just say "NSIDC." It is "University of Colorado" not Colorado University, but NSIDC is enough.

Response: Corrected as suggested.

Page 3, lines 23-24: Please provide a reference for these unrealistic buoy velocities. Are you referring here to Szanyi et al. (Geophys. Res. Lett., 43, doi:10.1002/2016GL069799, 2016)?

Response: The reference is presented in the revision (P4, L3-4).

Figure 1: a) Please give the lat/lon coordinates of each endpoint of the flux gate line. b)

There is no need for the compass rose, and in fact it is inappropriate on a map with the North Pole included. c) Is there a southern boundary of the PA? maybe at Bering Strait? This should be noted with a blue line, just like for the southern AA boundaries.

Response: a) the lat/lon coordinates of the endpoints of the fluxgate line are presented (see P5, L3-4). b) the compass rose is removed since it is not useful on a map with the North Pole. c) the southern boundary of PA is provided in the revised Figure, as marked with blue lines near the Bering Strait and Banks Island.

Equation 1: The units of F are length^2/time, yes? Please note this here.

Response: the unit of F is $km^2$/day and we note it in the text.

Table 1: Please provide the units for ice area flux. Also, it's "length" not width, I think (see comment above on consistency.)

Response: The units for sea ice area flux is provided and the "length" is used.

Page 5, line 17: Your English is generally quite good, but there are numerous minor errors in spelling and grammar. EG here it is "annual" not "annul."

Response: The manuscript is thoroughly read and examined by each author trying to avoid errors in spelling and grammar.

Bottom of page 5: Is it possible to provide a better estimate of the error in your calculations from the neglect of ice deformation (eg from numerical model studies)? Further, this error should be noted in the rest of your paper. EG if you find that there is a trend in total melt in the PA of X%, but this is smaller than the error from neglect of deformation, then do you have a significant result? Please discuss.

Response: At this point, we do not have a better estimate for the deformation contribution to sea ice loss. However, Lindsy et al(2009) provided an estimate due to deformation, which causes approximately 1% of sea ice area variation within the Atlantic side of the Arctic Ocean. Since the Pacific sector is dominated by ice divergence process, the sea ice coverage loss due to convergence may be less than that of the Atlantic sector. Thereby, we use 1% as a upper limit in sea ice changes due to deformation in the Pacific side, which is fairly less than the trend in total melt sea ice in PA (3.2%/yr for the period 1979-2016).

Page 6, lines 10-11: Is this DA index publicly available? Could you provide a link? AGU journals require a discussion of data availability; The Cryosphere may not, but it is all of our responsibility to discuss data availability so that our results can be independently reproduced.

Response: The DA index is not publicly available but a reference is provided (Wu et al., 2005). For the DA index data from 1979-present, it is kindly provided by Wu Bingyi through E-mail (bywu@fudan.edu.cn.).

Page 6, line 14: "precipitable"

Response: Corrected as suggested.

Figure 3: It is not immediately obvious to me that there is more blue on the left and more red on the right. Some nice further analysis is provided in subsequent plots, but for this one, I might suggest the addition of the annual mean anomaly on a new bottom row, which could more clearly summarize the trend.

Response: the addition of the annual mean anomaly is shown in the bottom row.

Page 9, line 2: This is very interesting. Any thoughts on why the extremes are not

changing?

Response: Despite the extremes are not changing overall, we further identify that the extreme low anomalies (A⩽ -1) reduce from 20.8% in P1 to 10.4%, while the extreme high anomalies (A > 1) increases from 7.5% to 16.6%. This shift in sea ice exchanges between the PA and the AA sectors may indicate a shift of atmospheric circulation toward a pattern facilitating sea ice export out of the PA side (Wu et al., 2005; Zhang et al., 2008; Jia et al., 2009). (Please see P10, L13-15).

Figure 8: Use same vertical scale for both panels. Note that the line goes from N. America (left) to Eurasia (right).

Response: the same vertical scale is used for both panels in the revised manuscript. Also, following the suggestion, the endpoint location of the lines is given in the caption of Figure 8. (see P14, L2-4)

Figure 9: It is nearly impossible for me to see these panels clearly and thus to interpret this figure. Could you try to make a better one, maybe with fewer panels?

Response: For a clear demonstration, we selected fewer panels with five-year estimates in each decade since 1980s being presented. Additionally, legend is given for easy identification of sea ice area changes due to advection or melting processes.

Page 13, lines 6-9: Are you saying that melting in the AA is of similar magnitude to that in the PA, but that the AA ice is getting replenished by PA ice? This would be a major new result if true. But I kind of doubt that it is true, given that the AA is (in the mean) farther north than the PA, and so probably there is more melt in the PA.

Response: We do not expect to express the idea that melting in AA has a similar magnitude as that in the PA, which is not real. The regarding sentences are rewritten or removed to avoid this misunderstanding. A notice is added to show that AA in higher latitudes is likely less subject to melting than that in PA. (see P14, L11-14)

Section 4 Discussion re climate indices: The Arctic community went through a phase in which everything was correlated with climate indices. This fad has faded as ice continues to decline independently of climate indices. Further, climate indices don't provide predictive skill. It is unclear to me what they do provide, specifically in the present context of this paper. I might suggest that you write some introductory words to this section that explain why you are correlating your results with these indices, and then at the end, what your significant correlations provide in terms of new insight.

Response: The explanatory words are presented in the first paragraph of Section 4 and a summary of the new quantitative findings is given in the last paragraph. In brief, wind forcing has been significant in modulating the sea ice variability in summer. The Arctic-wide wind forcing is linked to large-scale atmospheric circulation patterns. Hence, the connection between the sea ice area loss in the PA sector and three typical atmospheric indices (AO, NAO, and DA) are assessed here. The temporal changes of the correlation among different decades (Table 4) provides us the new evidence of a shifting atmospheric regime in the Arctic.

---

## Author Comment (AC3) · 9 Apr 2019

Response comments to RC1 are included in the attached .zip file. Note that the caption of the response is "Response-to-RC1.pdf".

Please also note the supplement to this comment:
https://www.the-cryosphere-discuss.net/tc-2019-11/tc-2019-11-AC3-supplement.zip
* * *

---

## Author Comment (AC4) · 9 Apr 2019

Response to RC2 is included in the attached ZIP file, entitled as "Reponse-to-RC2.pdf".

Please also note the supplement to this comment:
https://www.the-cryosphere-discuss.net/tc-2019-11/tc-2019-11-AC4-supplement.zip